# OC-space: a Unifying Perspective on Verification of Tree Ensembles

**Timo Martens** [1 2]  **Laurens Devos** [1 2]  **Lorenzo Cascioli** [1 2]  **Wannes Meert** [1 2]  **Hendrik Blockeel** [1 2]  **Jesse Davis** [1 2]

## Abstract

We study the problem of verifying whether certain properties such as robustness or fairness hold in an ensemble of decision trees. This problem is known to be NP-hard, with most research targeting a solution to a specific verification task. We explore the problem through the lens of an ensemble's OC-space: the set of all possible combinations of the individual trees' predictions. This provides a unifying view that yields a more generic and flexible approach to verification. We show that a wide variety of existing verification tasks can be (1) framed as simple searches through OC-space, and (2) answered in time linear or quadratic in the size of the OC-space. Moreover, the search can be made more efficient by using spatial index structures. Interestingly, while the OC-space can grow exponentially with the ensemble's size, in practice it is often feasible to enumerate all output configurations. Empirically, we show that our generic approach can be faster than approaches targeting a single verification task.

## 1. Introduction

Tree ensembles such as gradient boosted decision trees (Friedman, 2001) and random forests (Breiman, 2001) are a commonly used model class. Because they tend to offer strong predictive performance, they are often used to tackle significant real-world problems (Blockeel et al., 2023). Beyond predictive performance, many applications require that learned models exhibit properties such as robustness, fairness or safety. The goal of verification is to develop algorithms that can check (i.e., prove) whether or not a model satisfies a specific property of interest.

A considerable body of literature exists about developing verification approaches for ensembles of decision trees (Kantchelian et al., 2016; Chen et al., 2019; Andriushchenko & Hein, 2019; Ranzato & Zanella, 2020; Törnblom & Nadjm-Tehrani, 2020; Devos et al., 2021a;b). This is known to be a computationally challenging problem: verification tasks on additive ensembles are NP-complete (Kantchelian et al., 2016). Many existing approaches cope with this complexity by developing bespoke algorithms optimized towards a specific target task such as a specific definition of fairness (Calzavara et al., 2023) or robustness with respect to a specific distance function (Wang et al., 2020). But developing a new approach for each task is cumbersome. This raises the question whether a generic approach could handle a wider range of tasks equally well.

We start by obtaining novel insights into why the verification problem is computationally challenging. Intuitively, the issue arises from the fact that tree ensembles combine the predictions made by many trees, which is also the strength of the method from the standpoint of obtaining good predictive performance. Given an example, a prediction is made by sorting it to a leaf node in each tree, which yields an *output configuration*, i.e., the ordered set of leaf nodes – one from each tree in the ensemble – reached by the example. The output configuration fully determines the ensemble's resulting prediction. For this reason, verifying that a property holds for all possible instances (of which there are infinitely many) often reduces to verifying it for a finite number of output configurations (of which there are exponentially many).

Viewing verification through the prism of output configurations yields a number of interesting benefits, all of which we will demonstrate in this paper. First, it allows us to provide a unifying perspective on tree ensemble verification: we show that most tasks can be framed as a search over output configurations. While some special-purpose methods implicitly reason over output configurations (Chen et al., 2019; Wang et al., 2020; Törnblom & Nadjm-Tehrani, 2020; Devos et al., 2021b; Cascioli et al., 2024), no prior work has made this link explicit.

Second, the unifying view facilitates verification of novel properties for which no methods have yet been proposed. For example, instead of checking robustness for individual examples, one could globally check the maximum change in predicted class probability within a given perturbation bud-

[1] KU Leuven Department of Computer Science, Leuven, Belgium [2] Leuven.AI, KU Leuven Institute for Artificial Intelligence, Leuven, Belgium. Correspondence to: Timo Martens <timo.martens@kuleuven.be>.

*Proceedings of the 43rd International Conference on Machine Learning*, Seoul, South Korea. PMLR 306, 2026. Copyright 2026 by the author(s).

get on the input (i.e., reason about the slope of the model).

Third, precomputing and storing the OC-space allows for the design of very simple and generic verification algorithms. This gives significant flexibility as they are agnostic to, e.g., the distance function used in robustness checking. The algorithms are easily shown to have linear or quadratic time complexity in the size of the OC-space. Moreover, they can be sped up substantially by using spatial index structures. Storing and indexing the OC-space is a one-off cost that is amortized over subsequent verification tasks.

Empirically, we show our approach's breadth by using it to solve four different verification tasks, and its flexibility by considering a less typical distance metric in robustness checking. We find that the approach is practically feasible for a broad range of models: storing all output configurations is realistic for moderate-sized ensembles, and larger ensembles can often be compressed sufficiently (at a minimal cost in performance) to make verification feasible. This allows users to trade-off performance for verifiability. Practically, our approach offers similar or better (up to two orders of magnitude) runtime performance than existing verifiers that target a specific task.

## 2. Preliminaries

**Decision Trees**   A decision tree is a rooted tree that represents a function $T : \mathcal{X} \to \mathcal{Y}$. Each internal node of the tree is annotated with a function (called a test) that maps $\mathcal{X}$ onto a finite set of outcomes, each of which is associated with one child of the node. Furthermore, each leaf is annotated with a value from $\mathcal{Y}$. For any $x \in \mathcal{X}$, $T(x)$ is then defined as follows: starting at the root, repeatedly move $x$ to the child node indicated by the test, until a leaf $l$ is reached; the annotation of this leaf is $T(x)$. We denote the auxiliary function that maps $x$ to a leaf as $\text{leaf}(x)$.

For the purpose of this paper, we assume $\mathcal{X} = \mathbb{R}^F$ with $F$ the input dimensionality, a.k.a. number of features, and $\mathcal{Y} = \mathbb{R}$. We further assume binary trees with tests of the form $x_f < \tau$ where $x_f$ is the $f$'th component (a.k.a. feature) of $x$ and $\tau$ some threshold value. This is consistent with most implementations of tree ensembles. Under these conditions, the subset of $\mathcal{X}$ for which $\text{leaf}(x) = l$ is of the form $\bigwedge_f v_f \le x_f < u_f$ with $v_f$ and $u_f$ lower and upper bounds on $x_f$ (possibly $\pm\infty$). We call this set the leaf's *box*, and denote it $\text{box}(l)$, or $\text{box}(x)$ for any $x \in \text{box}(l)$.

**Tree Ensembles**   Tree ensembles include popular algorithms such as gradient boosted decision trees (GBDTs) (Friedman, 2001) (e.g., XGBoost (Chen & Guestrin, 2016)) and random forests (Breiman, 2001). A tree ensemble $\boldsymbol{T}$ is a model that contains a set of decision trees $T_m$, $m = 1, \ldots, M$ and for any $x$ combines their predictions

$T_m(x)$ into a single prediction $\boldsymbol{T}(x)$.

We focus on gradient boosting. Because we consider a wide range of properties and both regression and classification tasks, we distinguish among three output forms of the ensemble. The raw scores $\boldsymbol{T}^{\text{raw}}(x) = \nu_0 + \sum_{m=1}^M T_m(x)$ are simply the sum of the predictions of the trees plus a constant base score $\nu_0$; this output is relevant for both classification and regression. In the context of classification, the raw scores can be converted into class probabilities $\boldsymbol{T}^{\text{prob}}(x)$ and then into hard labels $\boldsymbol{T}^{\text{label}}(x) = \arg\max_y \boldsymbol{T}_y^{\text{prob}}(x)$, with $\boldsymbol{T}_y^{\text{prob}}(x)$ the probability of class $y$.

**Verification**   In a strict sense, verification refers to checking whether a model satisfies some given property or not. The property is often formulated as a constraint that $\boldsymbol{T}$ must satisfy. Global constraints must hold for all $x \in \mathcal{X}$ or pairs $(x, x') \in \mathcal{X}^2$ whereas local constraints must only hold for a given $x$.[1]

In the broader sense, verification is often used not just to check a constraint but to answer related queries, such as: count how many instances violate some constraint, return the set of all violating instances, etc. In this paper, we use the term verification in this broader sense.

## 3. Output Configurations and OC-space

**Output Configurations.**   Given a tree ensemble $\boldsymbol{T}$, an output configuration (OC) (Devos et al., 2023) is a tuple $oc = (l^{(1)}, \ldots, l^{(M)})$ consisting of exactly one leaf per tree $T_m$ such that there exists an $x \in \mathcal{X}$ that visits those leaves, i.e., $l^{(m)} = \text{leaf}_m(x)$, with $m$ indexing tree $T_m$ in the ensemble. By definition, it holds that $\bigcap_m \text{box}(l^{(m)}) \ne \emptyset$ because otherwise no such $x$ would exist. Analogous to a leaf's box, an $oc$'s box is the subset of $\mathcal{X}$ that visits the leaves in the OC: $\text{box}(oc) = \bigcap_m \text{box}(l^{(m)}) = \prod_{1 \le f \le F} [v_f, u_f]$, where each $[v_f, u_f]$ is an interval constraining the $f$th feature. $\forall x \in \text{box}(oc)$, the ensemble predicts the same value.

An ensemble's *OC-space* is the set of all possible output configurations. Another way to think about the *OC-space* is as the set of all possible $\boldsymbol{T}^{\text{raw}}(x)$ that an ensemble can return. From a spatial perspective, the boxes associated with each output configuration form a partition of the input space with each part (i.e., box) associated with exactly one output produced by the ensemble. Figure 1 shows the OC-space for a small example ensemble.

Slightly abusing notation and terminology, we use the symbol $\mathcal{O}$ and term OC-space for both the set of output configurations and the set of corresponding boxes. Given example

---

[1]Note that this definition of global constraints is stricter than that in some earlier work (Leino et al., 2021), where global refers to a finite data set rather than the whole instance space.

(a)

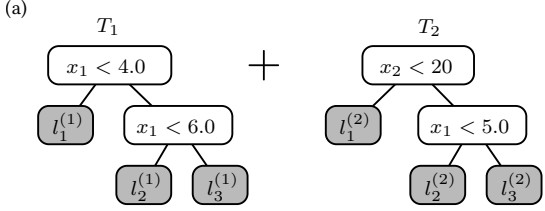

(b)

*Figure 1.* (a) An example ensemble of two trees. (b) A visual depiction of the ensemble's OC-space which partitions $\mathcal{X}$ into boxes. Each box is annotated with its OC. The size of the *OC-space* is 7. The leaf bound equals $3 \cdot 3 = 9$ and the feature bound equals $4 \cdot 2 = 8$. The pairs $(l_1^{(1)}, l_3^{(2)})$ and $(l_3^{(1)}, l_2^{(2)})$ are not OCs because of conflicting constraints on $x_1$.

$x$, we systematically use $b$ for the box associated with its output configuration and $y$ for the ensemble's output for $b$.

**Size of the OC-space**   Some leaf combinations are not an output configuration because of conflicting constraints on the feature values (the OC's box would be empty). For example, in Figure 1, $(l_1^{(1)}, l_3^{(2)})$ is not an OC because $x_1$ cannot be $< 4.0$ and $\geq 5.0$ simultaneously. Hence, the OC-space is not just the Cartesian product of all leaves. Therefore, the exact size of the OC-space of a given ensemble is difficult to determine except through enumeration. The following theorems give upper bounds on the size of the OC-space.

**Theorem 3.1** (Leaf Bound). $|\mathcal{O}| \leq \prod_m L_m$ *with $L_m$ the number of leaves of tree $T_m$.*

**Theorem 3.2** (Feature Bound). $|\mathcal{O}| \leq \prod_f (C_f + 1)$ *with $C_f$ the number of constants with which $x_f$ is compared in the ensemble.*

See Appendix A.1 for proofs. Because the number of trees can be chosen independently from the number of times a feature appears in a tree, the bounds are highly complementary: each can be orders of magnitude smaller than the other. Both bounds reflect worst-case scenarios: in practice, $|\mathcal{O}|$ may be much smaller than the bound suggests. Figure 6 in Appendix B shows the bounds' behavior on a few datasets.

## 4. Verification in OC-space

The main challenge in verifying tree ensembles is that they can represent very complex functions that are hard to reason

about. At the core of our approach lies the insight that within a single OC, $\boldsymbol{T}^{\mathrm{raw}}(x)$ (and hence $\boldsymbol{T}^{\mathrm{prob}}(x)$ and $\boldsymbol{T}^{\mathrm{label}}(x)$) is constant, and this decouples the complexity of verification from the function's complexity. Consequently, we can frame verification as a search over $\mathcal{O}$. More importantly, it allows us to provide a unifying perspective on verification for tree ensembles: most tasks considered in the literature can be solved by (a) enumerating and storing the full OC-space, and (b) then searching through $\mathcal{O}$ to check if the property holds. The search in step (b) is task specific with the following three high-level search patterns arising:

1. **all OCs** $b' \in \mathcal{O}$,

2. **all pairs of OCs** $b, b' \in \mathcal{O}$,

3. **pairs of OCs, where each OC comes from a different ensemble**, e.g., considering all OCs $b, b' \in \mathcal{O} \times \mathcal{O}'$ with $\mathcal{O}$ ($\mathcal{O}'$) the OC-space of ensemble $\boldsymbol{T}$ ($\boldsymbol{T}'$).

Viewing verification as operating on the OC-space has two other important advantages. First, the complexity of verification, as is obvious from the search patterns above, can be done in time polynomial in $|\mathcal{O}|$. Second, it allows specifying simple, generic algorithms that do not require to, e.g., commit to a specific distance function. This gives substantial flexibility which contrasts to existing approaches for, e.g., robustness checking which often are designed to work with a specific (family of) metric(s) (Devos et al., 2024; Kantchelian et al., 2016; Chen et al., 2019; Wang et al., 2020).

Table 1 lists a number of constraints or properties for which verification has been studied in the literature plus some new ones we propose, and shows how they can be checked in OC-space (Appendix C gives full pseudo code). The columns give the property name, a formal description of the property to be verified, and how it can be computed. For consistency and conciseness, all algorithms are shown as an aggregation function computed over a set $((\arg) \max, (\arg) \min, \Sigma, \mathrm{empty})$, where $\mathrm{empty}(S)$ returns true if and only if $S = \emptyset$). The algorithms rely on the following terminology and auxiliary functions. We call a box $b$ $f$-independent when $v_f = -\infty$ and $u_f = +\infty$ (i.e., whether $x$ is in $b$ or not does not depend on $x_f$), and $f$-dependent otherwise. We denote the subset of $\mathcal{O}$ that contains all $f$-dependent boxes as $\mathcal{O}_f$. Given a box $b$, $\mathrm{relax}(b, f)$ is the $f$-independent box obtained by setting $v_f = -\infty$ and $u_f = +\infty$ in $b$. We assume a distance metric $\delta$ is defined over the input space (e.g., the $L_1$, $L_2$ or $L_\infty$ norm). We define $\mathrm{mindist}(x, b') = \min_{x' \in b'} \delta(x, x')$ and $\mathrm{mindist}(b, b') = \min_{x \in b, x' \in b'} \delta(x, x')$. For bounded $b$, we define $vol(b) = \prod_f (u_f - l_f)$.

We now briefly discuss each considered property. These are grouped together according to which of the three identified

search patterns the verification algorithm employs.

## 4.1. Checking each OC

The following verification tasks require looking at each OC and hence are $O(|\mathcal{O}|)$:

**Plausibility of range** This requires that for all examples $x$, the model always returns a prediction that falls within a given range (Törnblom & Nadjm-Tehrani, 2020). This can be checked by iterating over all $b$ and checking if its corresponding $y$ falls within the range.

**Empirical robustness** Robustness relates to how easily a model's prediction can be changed by making small perturbations to the input. It is typically checked locally, for a given $x$. We distinguish three different queries that could be asked about $x$: (1) for a given $\epsilon$, check that no example $x'$ exists with $\delta(x, x') < \epsilon$ and $y' \neq y$ (Leino et al., 2021); (2) what is the smallest $\epsilon$ for which there is an $x'$ within distance $\epsilon$ of $x$ such that $y' \neq y$? (Kantchelian et al., 2016); (3) among all $x'$ within distance $\epsilon$ from $x$, which one maximizes $|y - y'|$ (Devos et al., 2021b)? The first two questions are specific to classification, but can easily be reformulated for regression by changing $y' \neq y$ into a threshold condition on $|y - y'|$.[2]

The algorithms in Table 1 formulate the questions into set operations on $\mathcal{O}$. All algorithms require performing a linear scan over $\mathcal{O}$. The algorithms are agnostic to the chosen distance function, so one can easily impose combinations of constraints such as putting a budget on (a) the number of perturbed features, (b) the maximum perturbation allowed per feature, and (c) the total amount of perturbation.

## 4.2. Checking pairs of OCs

Another family of verification tasks are characterized by the need to compare (all) pairs of OCs and hence are $O(|\mathcal{O}|^2)$:

**Average slope** Robustness is typically tested locally, for a given $x$. Its global counterpart (does a non-robust $x$ exist?) is not useful for classification ensembles because instances on the decision surface are by definition not robust.

A global notion of robustness does become possible if we express it in terms of $\boldsymbol{T}$'s average slope over some distance. The following *average-slope* constraint states that $y$ cannot increase by more than $\Delta_y$ over a distance less than $\Delta_x$:

$$\delta(x, x') \leq \Delta_x \Rightarrow |y - y'| \leq \Delta_y.$$

Again, this can be checked efficiently in OC-space because $y$ is constant within each box. A pair $x, x'$ violating the con-

straint exists if and only if a pair of boxes $b, b'$ is found with $\text{mindist}(b, b') \leq \Delta_x$ while $|y - y'| > \Delta_y$. The average-slope constraint is applicable to both regression and classification ensembles. In the latter case, one can check whether small changes in the input only yield small changes in the predicted probability associated with each class. To our knowledge, this average-slope constraint is novel for tree ensembles.

**OC-level robustness** Whereas empirical robustness is a data-dependent measure (it is computed using a finite set of examples), thinking on the level of OCs allows us to propose a novel data-independent notion of robustness for classification problems. Specifically, for each $b \in \mathcal{O}$, we can check if $\nexists b'$ s.t. $\delta(b, b') < \epsilon$ and $y \neq y'$. If true, we have verified that for all examples $x \in b$, no adversarial example exists within distance $\epsilon$. Lifting this to the global level, for a bounded instance space, one can compute the percentage of the volume of $\mathcal{X}$ that is robust. As explained previously, the existence of a decision boundary precludes being globally robust. Hence, this percentage will always be $< 1$.

**Fairness** We use the (lack of) causal discrimination as our fairness definition (Calzavara et al., 2023). This means that for tree ensemble $\boldsymbol{T}$, we consider fairness with respect to a protected attribute $p$.[3] Specifically, this requires checking whether there exist two examples $x, x' \in \mathcal{X}$ such that (1) $x_i = x_i'$ for all attributes $i \neq p$, (2) $x_p \neq x_p'$, and (3) $\boldsymbol{T}^{\text{label}}(x) \neq \boldsymbol{T}^{\text{label}}(x')$.

Because all instances that have the same output configuration must have the same predicted label, we can reason over output configurations to verify fairness. We essentially need to check for each $p$-dependent box $b$ whether $b$ contains any $x$ for which changing $x_p$ moves $x$ to a different box $b'$ with $y' \neq y$. The box $\text{relax}(b, p)$ contains all $x'$ that such an $x$ could move to; hence, if any $b'$ with $y' \neq y$ overlaps with $\text{relax}(b, p)$, the fairness condition requirement is violated. For a formal proof of correctness, see Appendix D.

**Monotonicity** The constraint $\forall x, x' : x_m \leq x_m' \wedge (\forall f \neq m : x_f = x_f') \Rightarrow y \leq y'$ (i.e., $y$ must monotonically increase with $x_m$, for a given $m$) is violated if a pair $x, x'$ can be found such that $x_m \leq x_m' \wedge (\forall f \neq m : x_f = x_f') \wedge y > y'$. This check is very similar to the fairness check: instead of "changing $x_p$ should not change $y$", we now have "increasing $x_m$ should not decrease $y$", i.e., $l_m < u_m'$ (so $b'$ contains an $x'$ with $x_m \leq x_m'$) and $y' < y$ indicates a violation.

---

[2]Note that these queries typically only consider correctly classified examples.

[3]This can also be a set of protected attributes. For a simpler representation we show the proof with a single attribute but the results generalize as there is an attribute transformation possible.

*Table 1.* Overview of verifiable properties. We use $v, u$ to denote the lower respectively upper bound of a box.

| Type | Constraint / property | Algorithm |
|---|---|---|
| Plausibility of range | $\forall x \in \mathcal{X} : c_1 \leq T(x) \leq c_2, c_1, c_2 \in \mathbb{R}$ | $\mathrm{empty}\{b \in \mathcal{O} \mid y < c_1 \vee y > c_2\}$ |
| Empirical robustness | for a given $x \in \mathcal{X}$ and (1,3) $\epsilon > 0$: 
 (1) $\nexists x' : \delta(x, x') < \epsilon \wedge \boldsymbol{T}(x') \neq \boldsymbol{T}(x)$ 
 (2) smallest $\epsilon$ s.t. $\exists x' : \delta(x, x') \leq \epsilon \wedge \boldsymbol{T}(x') \neq \boldsymbol{T}(x)$ 
 (3) $\arg\max_{x', \delta(x', x) < \epsilon} \|\boldsymbol{T}(x) - \boldsymbol{T}(x')\|$ | (1) $\mathrm{empty}\{b' \in \mathcal{O} \mid \mathrm{mindist}(x, b') < \epsilon \wedge y' \neq y\}$ 
 (2) $\min\{\mathrm{mindist}(x, b') \mid b' \in \mathcal{O} \wedge y' \neq y\}$ 
 (3) any $x^* \in b^*$ with $\delta(x, x^*) < \epsilon$ where 
 $b^* = \arg\max_{b' \in \mathcal{O}, \mathrm{mindist}(x, b') < \epsilon} \|y - y'\|$ |
| Average Slope | $\forall x, x' : \delta(x, x') \leq \Delta_X \Rightarrow \|\boldsymbol{T}(x) - \boldsymbol{T}(x')\| \leq \Delta_y$ | $\mathrm{empty}\{(b, b') \in \mathcal{O} \mid \mathrm{mindist}(b, b') \leq \Delta_x$ 
 $\ldots \qquad\qquad \wedge \|y - y'\| > \Delta_y\}$ |
| OC-level robustness | $\dfrac{vol(\{b \mid \nexists b' \in \mathcal{O} : y' \neq y \wedge \mathrm{mindist}(b, b') < \epsilon\})}{vol(\mathcal{X})}$ | $\dfrac{\sum_{b : \nexists b' \in \mathcal{O} : y' \neq y \wedge \mathrm{mindist}(b, b') < \epsilon} vol(b)}{vol(\mathcal{X})}$ |
| Fairness | Checking global fairness over $\mathcal{X}$: 
 $\forall x, x' \in \mathcal{X} : (\forall i \neq p : x_i = x_i') \Rightarrow \boldsymbol{T}(x) = \boldsymbol{T}(x')$ 

 Part of $\mathcal{X}$ where model is unfair: 
 $\{x \mid \exists x' : (x, x') \text{ violates fairness}\}$ | $\mathrm{empty}\{(b, b') \in \mathcal{O}_p^2 \mid y' \neq y$ 
 $\ldots \qquad\qquad \wedge \mathrm{relax}(b, p) \cap b' \neq \emptyset\}$ 

 $\{\mathrm{relax}(b, p) \cap b' \mid (b, b') \in \mathcal{O}_p^2 \wedge y' \neq y\}$ |
| Monotonicity | $\forall x, x' \in \mathcal{X} : (\forall i \neq p : x_i = x_i') \wedge x_m < x_m' \Rightarrow$ 
 $\boldsymbol{T}(x) \leq \boldsymbol{T}(x')$ | $\mathrm{empty}\{(b, b') \in \mathcal{O}_m^2 \mid y' < y \wedge u_m' > l_m$ 
 $\ldots \qquad\qquad \wedge \mathrm{relax}(b, p) \cap b' \neq \emptyset\}$ |
| Safety | $\max_{x \in \mathcal{X}} \|\boldsymbol{T}(x) - \boldsymbol{T}_{\mathrm{ref}}(x)\|$ | $\max\{\|y - y'\| \mid (b, b') \in \mathcal{O} \times \mathcal{O}_{\mathrm{ref}} \wedge b \cap b' \neq \emptyset\}$ |
| Multiplicity | volume in $\mathcal{X}$ where $\|\boldsymbol{T}_1(x) - \boldsymbol{T}_2(x)\| > \epsilon$ | $\sum\{vol(b \cap b') \mid (b, b') \in \mathcal{O}_1 \times \mathcal{O}_2 \wedge \|y - y'\| > \epsilon\}$ |

## 4.3. Comparing OCs from different models

Some verification tasks require comparing different models. Such properties can also be checked in OC-space and scale $O(\prod_{i=1}^{ms} \|\mathcal{O}_i\|)$ with $ms$ the number of models considered.

**Safety** Wei et al. (2022) assess the safety of a new model $\boldsymbol{T}$ by comparing it to a previous (e.g., extensively tested) model $\boldsymbol{T}_{\mathrm{ref}}$ and computing the maximum deviations between $\boldsymbol{T}$ and $\boldsymbol{T}_{\mathrm{ref}}$: $\max_{x \in \mathcal{X}} \|\boldsymbol{T}(x) - \boldsymbol{T}_{\mathrm{ref}}(x)\|$. They do this for a single-tree $\boldsymbol{T}_{\mathrm{ref}}$. We here consider the case where both $\boldsymbol{T}$ and $\boldsymbol{T}_{\mathrm{ref}}$ are ensembles with different OC-spaces $\mathcal{O}, \mathcal{O}_{\mathrm{ref}}$. The algorithm answers the query in time $O(\|\mathcal{O}\| \cdot \|\mathcal{O}_{\mathrm{ref}}\|)$.

**Multiplicity** Watson-Daniels et al. (2023) study *predictive multiplicity*: variability among predictions of models with near-optimal global performance (i.e., all models whose performances falls within a specified margin of the best one). They quantify discrepancy between models as the number of training instances on which their predictions differ by more than $\epsilon$. The algorithm in Table 1 quantifies discrepancy with respect to the whole input space rather than a dataset, by returning the total volume where $y$ and $y'$ differ. (This assumes a bounded input space $\mathcal{X}$).

## 5. Efficiently Searching the OC-space

The only assumption made by the Algorithms listed in Table 1 is that the OC-space can be enumerated. Conceptually, these can be straightforwardly implemented by storing the OC-space in a list and answering the queries using (nested) for-loops. However, faster algorithms can be devised by indexing the OC-space.

Since the stored objects are essentially hyperrectangles in the instance space, spatial index structures are a natural choice. Among these, R*-trees (Beckmann et al., 1990) and X-trees (Berchtold et al., 1996) are the most relevant because they allow the stored objects to be hyperrectangles, not just individual vectors. These spatial indexes work by creating bounding boxes around the stored objects. At each level of the tree-structured index, each node contains a bounding box, and to look up an object, one needs to only explore the nodes (and their subtrees) whose bounding box contains it. The index works best when at each level only one bounding box can possibly contain the sought object. This ideal is not always achievable, for instance (among other reasons) because the stored boxes may themselves overlap. The performance of R*-trees is known to degrade in high-dimensional spaces, due to excessive overlap among bounding boxes. X-trees compensate for this by creating linear structures when the R*-tree structure is expected to suffer from this.

In OC-spaces, boxes cannot overlap with each other, and non-overlapping bounding boxes always exist. Interestingly, it is not necessary to search for them, because OC-spaces have an interesting property that gives us the bounding boxes for free.

To see this, first recall the following concepts. A *partition* of a space $\mathcal{X}$ is a set of non-empty subsets $P = \{S_1, \ldots, S_p\}$ such that each element of $\mathcal{X}$ occurs in exactly one of the $S_i$. A partition $P = \{S_i\}_i$ *subpartitions* a partition $P' = \{S'_j\}$ if each $S_i$ is a subset of exactly one $S'_j$. This implies that each $S'_j$ equals the union of one or more $S_i$. For instance, $\{\{1, 2\}, \{3\}, \{4\}, \{5\}\}$ subpartitions $\{\{1, 2, 3\}, \{4, 5\}\}$, while $\{\{1, 2\}, \{3, 4, 5\}\}$ does not.

We call $\boldsymbol{T}'$ a *sub-ensemble* of $\boldsymbol{T}$ when it can be obtained by dropping and/or pruning some trees in $\boldsymbol{T}$. More precisely: $\boldsymbol{T}' = \{T'_1, \ldots, T'_{M'}\}$ is a sub-ensemble of $\boldsymbol{T} = \{T_1, \ldots, T_M\}$ if an injective mapping $g : \boldsymbol{T}' \to \boldsymbol{T}$ exists such that each $T' \in \boldsymbol{T}'$ is a rooted subtree of $T = g(T')$. $T'$ is a rooted subtree of $T$ if it has the same root and internal nodes as $T$, except that some internal nodes of $T$ are turned into leaves (and the nodes below it are dropped). The content of the leaves of $T'$ does not matter.

We now have the following theorem (proof in Appendix A.2).

**Theorem 5.1.** *When $\boldsymbol{T}'$ is a subensemble of $\boldsymbol{T}$, the OC-space of $\boldsymbol{T}$ subpartitions the OC-space of $\boldsymbol{T}'$.*

In other words, for any subensemble $\boldsymbol{T}'$ of $\boldsymbol{T}$, the boxes in $\mathcal{O}_{\boldsymbol{T}'}$ are bounding boxes for one or more boxes in $\mathcal{O}_{\boldsymbol{T}}$. These bounding boxes do not overlap (because the boxes of an OC-space cannot overlap). It is therefore possible to build an index structure, similar to an R*-tree, by simply considering subensembles of $\boldsymbol{T}$.

In this paper, we consider two such index structures. The **Rootbox index** is a one-level index that uses as bounding boxes the OC-space obtained when all trees in $\boldsymbol{T}$ are pruned to their root; that is, each tree contains only a root and two leaves. Because of Theorems 3.1 and 3.2, this index has at most $\min(2^M, \prod_f(C_f + 1)) = \prod_i(C_f + 1)$ entries[4], with $C_f$ the number of times feature $x_f$ occurs in any root.

The second index structure we consider is the **OC-Tree**. It is a multi-level index that uses at level $k$ the OC-space of the first $k$ trees from the ensemble, $\boldsymbol{T}'_k = \{T_1, \ldots, T_k\}$. As $\boldsymbol{T}'_k$ is a subensemble of $\boldsymbol{T}'_{k+1}$, level $k$ contains bounding boxes for level $k + 1$. The size of this index grows exponentially with $k$ but as long as its depth is less than $M$, the number of entries is smaller than the OC-space being indexed.

These index structures can tremendously speed up queries that find all boxes that overlap with a given box (e.g., fair-

---

[4]Because $\sum_f C_f = M$, $\prod_f(C_f + 1)$ cannot exceed $2^M$.

ness queries) or are within a given distance from it (e.g., robustness queries). These conditions are monotonic: when a box satisfies the condition, so does any box containing it. Conversely, when a bounding box of an index entry fails the condition, all boxes inside it must fail. More formally, we have the following theorem (proof in Appendix A.2).

**Theorem 5.2.** *If the distance from example $x$ to bounding box $B$ exceeds $d$, then the distance from $x$ to any box $b$ that lies entirely inside $B$ must also exceed $d$.*

Our indexes utilize this theorem to speed up verification. When searching for boxes within a given distance $d^*$ to $x$, we avoid enumerating boxes inside any bounding box $B$ whose distance to $x$ exceeds $d^*$. This often makes quadratic queries close to linear in practice. Below we give two examples for properties discussed in Table 1 (pseudocode is given in Appendix C).

**Example 1.** In case that we want to find (the distance to) the nearest adversarial example ($d^*$) of $x$, $d^*$ is constantly being updated. At every step in the iteration process, we do not consider any bounding box $B$ that is further from $x$ than the smallest distance found until now. Consequently, as $d^*$ decreases, increasingly more bounding boxes can be pruned away.

**Example 2.** For the average slope constraint, we rely again on the same principle, but every bounding box also keeps track of the $y$-range of the boxes $b \in B$. The average slope from an example $(x, y)$ to any $(x', y') \in B$ is $|y' - y|/\delta(x, x')$. This cannot exceed $\max(y - m, M - y)/\mathrm{mindist}(x, B)$, with $m$ and $M$ respectively the smallest and largest value of $y'$ found in $B$, because $|y' - y|$ cannot exceed the numerator and $\delta(x, x')$ is at least $\mathrm{mindist}(x, B)$. Hence, similarly to Example 1, all boxes in a bounding box $B$ with $\max(y - m, M - y)/\mathrm{mindist}(x, B) \leq c$ (with $c$ the greatest slope found until now) can be pruned away when looking for the smallest global upper bound on the slope.

## 6. Related Work

Some existing tree ensemble verification approaches operate in something akin to OC-space (Chen et al., 2019; Wang et al., 2020; Törnblom & Nadjm-Tehrani, 2020; Devos et al., 2021b; Cascioli et al., 2024). However, these approaches typically pursue a search-based approach and only generate the output configurations needed to answer a specific query.

Another approaches to verifying tree ensembles exploit the logical structure of decision trees (Cui et al., 2015; Kanamori et al., 2020; Calzavara et al., 2020; Ranzato & Zanella, 2020; Parmentier & Vidal, 2021; Murtovi et al., 2023). These approaches compile the ensembles and verification queries into a mathematical language and use an off-the-shelf solver to answer the query. Thus, these approaches operate on the model as opposed to working in OC-space.

Notable examples in this regard are from Kantchelian (2016) who work with a MILP encoding and several approaches based on SMT encodings (Devos et al., 2021a; Sato et al., 2020; Einziger et al., 2019).

Regardless of algorithmic approach, most existing tree ensemble verification methods are narrowly focused: most considers either robustness verification (Chen et al., 2019) or adversarial example generation (Zhang et al., 2020). Other works target a specific tasks, such as fairness (Calzavara et al., 2023). Some work (Devos et al., 2021a;b) can check multiple properties, but these papers do not cover the same breadth of tasks discussed here.

Beyond tree ensembles, there is also a large body of literature on verification for other model classes such as (deep) neural networks (Katz et al., 2017; 2019; Henriksen & Lomuscio, 2021; Liu et al., 2021) and probabilistic models (Kwiatkowska et al., 2011).

# 7. Experiments

We experimentally validate three crucial claims:

1. In many practical scenarios, enumerating and storing the OC-space is feasible.

2. Verification in OC-space is often as efficient as special-purpose methods.

3. Verification in OC-space offers more flexibility than special-purpose algorithms.

For this purpose, we have trained XGBoost models on 16 datasets (details in Appendix E, Table 3) using a grid over the following hyperparameters: number of trees $\in [10, 25, 50, 100]$, maximum tree depth $\in [4, 6, 8]$ and learning rate $\in [0.1, 0.25, 0.5, 1.0]$, using 5-fold cross-validation. Standard ensemble learners such as XGBoost struggle to learn small yet performant models. Often, it is possible to compress ensembles without harming predictive performance (Devos et al., 2025; Liu & Mazumder, 2023). Because ensemble size is relevant for our approach, we compress all learned XGBoost models using LOP (Devos et al., 2025), constraining the compressed models to have balanced accuracy within half a percentage point of the original XGBoost model on the validation set. For each dataset, we consider all compressed models on the Pareto front (i.e., no other compressed ensemble has both better balanced accuracy and fewer leaves).

Within each experiment, all compared methods[5] were executed on the same machine, single-threaded and having access to 100GB of memory.

---

[5]The code can be found at https://github.com/ML-KULeuven/OC-space

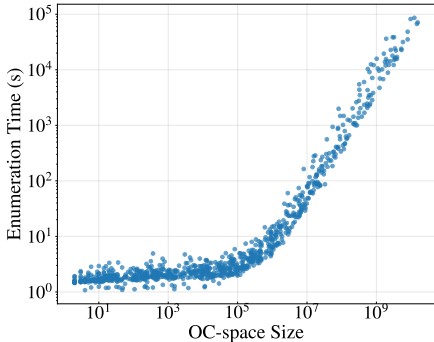

*Figure 2.* The time in seconds needed to enumerate the OC-space as a function of $|\mathcal{O}|$ for all models on the Pareto fronts for each considered dataset. Only models that could be enumerated within the time limit of 24 hours are shown.

Reported timings for queries exclude the time needed for enumerating the OC-space and building the index, which are fixed one-time costs that are amortized over many subsequent queries.

## 7.1. Enumerating and indexing the OC-space

Considering all folds over the 16 datasets, there are 1065 models in total on the Pareto front. We tried to enumerate the OC-space of these applying a wall-clock time limit of 24h, writing the OC-space to disk. In total, we were able to enumerate 82.8% of all considered models. This includes models with up to 67 trees and tree depths up to 8. For 9 out of 16 datasets, all models could be enumerated. Figure 2 plots the runtime as a function of $|\mathcal{O}|$ for all enumerated models across all datasets. 82.1% of all enumerations take less than 1 minute and 88.3% less than 5 minutes.

Our current implementation of the indexes supports only main memory storage. Under this constraint, index construction is possible for 90.2% of the enumerable models. 95.8% of the Rootbox indexes and 91.4% of the OC-Tree indexes are built in less than 1 minute, and 99.4% and 97.1%, respectively, in less than 5 minutes. This underscores the practical feasibility of enumeration and index construction. Appendix E.1 contains more detailed results on enumeration and index construction.

## 7.2. Comparison against special-purpose approaches

**Adversarial robustness**  We compare our proposed approach to Kantchelian et al.'s MILP approach (Kantchelian et al., 2016) for finding nearest adversarial examples, as it is the reference method for exact verification.[6] Specifically, we select 500 correctly classified test examples and

---

[6]We use the public implementation available at https://github.com/laudv/veritas and use the same configuration for Gurobi.

*Table 2.* The number of empirical robustness verification queries required to amortize the cost of enumerating and indexing the OC-Space compared to always using Kantchelian et al.'s MILP approach. Results are reported for both a Rootbox and OC-Tree index on each dataset where all models could be enumerated within 24 hours. Values are computed as described in Appendix E.2.

| Dataset | Rootbox | OC-Tree |
|---------|---------|---------|
| Adult | 1022 | 1830 |
| California | 2175 | 4892 |
| Compas | 1537 | 8492 |
| Credit | 726 | 903 |
| DryBean | 1006 | 829 |
| Phoneme | 869 | 718 |
| Spambase | 4146 | 26437 |
| Vehicle | 2857 | 2331 |
| Volkert | 3244 | 2191 |

try to perform an evasion attack: for each instance $x$ we find the smallest $\epsilon$ for which there exists an $x'$ such that $||x - x'||_\infty < \epsilon$ and $y \neq y'$.

We investigate the use of three indexes: a linear scan that uses separate lists for positive and negative OCs, a Rootbox index and an OC-Tree index with depth $\max(M - 2, 10)$. For the latter two, queries are answered by first linearly scanning the OCs in the instance's smallest bounding box to find a first value for $\epsilon$. After this, the full index is searched, pruning all bounding boxes farther from $x$ than the smallest $\epsilon$ found until then.

Figure 3 shows the average time required to perform an evasion attack for the different approaches on two datasets (Figure 8 and Table 6 in Appendix E show details for all datasets). Using the indexes, finding the closest adversarial example is typically 10-100 times faster than using Kantchelian et al.'s (2016) MILP approach. Unsurprisingly, performing a linear scan scales much worse than any other approach.

Enumerating and indexing the OC-space is a fixed cost and it only makes sense to incur this time cost if a sufficient number of subsequent queries must be answered. Table 2 shows the number of queries after which the cumulative run time of Kantchelian et al.'s approach is greater than the cumulative run time for using a Rootbox index or an OC-Tree index (i.e., including the time needed to enumerate and index the OC-space). The results show that this crossover point is typically reached after a few hundred to a few thousand queries. It usually takes longer to build an OC-Tree index because it is more fine-grained than the Rootbox index. For both indexes, the overhead is negligible when robustness will be checked for each prediction made by a model or on a large set of examples as part of an offline robustness evaluation (e.g., prior to model deployment).

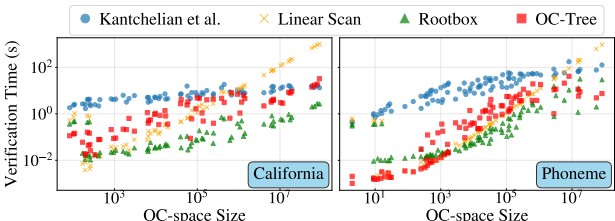

*Figure 3.* The average time in seconds to find the nearest adversarial example (empirical robustness definition 2) as a function of the size of the OC-space using Kantchelian et al.'s MILP approach, a Linear Scan of the OC-space, a Rootbox index, and an OC-Tree index for the California (left) and Phoneme (right) datasets. The average is computed over 500 correctly classified test examples.

**Fairness**   We use Calzavara et al.'s (2023) (lack of) causal discrimination definition for testing fairness on the Adult and Compas datasets, using Sex as the protected attribute. Specifically, for each model on the Pareto front, we enumerate all boxes that contain instances for which changing the protected attribute changes the prediction. We compare our approach, using the OC-Tree index, with Veritas (Devos et al., 2021b), which uses a search-based approach that operates on the ensemble itself during verification. Similar to our approach, Veritas returns exact results for this setting and can enumerate all unfair boxes.

Figure 4 compares the runtimes for these two approaches. The OC-Tree search approach is faster in 36/70 models on Adult, and 44/55 models on Compas. Appendix E.2 shows more detailed results.

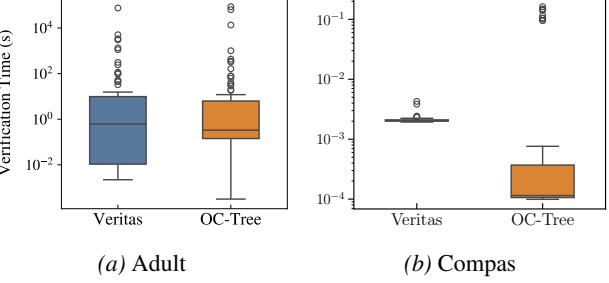

*(a)* Adult          *(b)* Compas

*Figure 4.* Distribution of time in seconds to verify (the lack of) causal discrimination (fairness) for all models on the Pareto front for the Adult (left) and Compas (right) datasets using Veritas and a OC-Tree.

### 7.3. Flexibility in the verifiable properties

**Different distance norms**   Most methods designed to find adversarial examples exclusively use $|| \cdot ||_p$ as a distance metric (Kantchelian et al., 2016; Devos et al., 2021b; Chen et al., 2019). However, in several scenarios this might be unrealistic (Wang et al., 2020). We therefore experiment with a more general evasion attack using a hybrid distance function that combines two different norms. Specifically, a

valid adversarial example $x'$ must satisfy (1) a per-attribute perturbation budget $||x - x'||_\infty < \epsilon$ and (2) a total budget $||x - x'||_1 < B$. In other words, there is a limit on how much each feature can be modified, as well as on the aggregated sum of the perturbations.

We analyze the use of a Rootbox index similar as above where any OC falling in a bounding box $b$ with $||b - x||_\infty > \delta$ and $||b - x||_1 > B$ can be ignored during the search process. This is compared against an SMT implementation from Devos et al. (2021a), where this setting was originally introduced. Figure 5 shows the timings for the two methods. The median time for searching the indexed OC-space is a factor 100-1000 faster than the SMT approach.

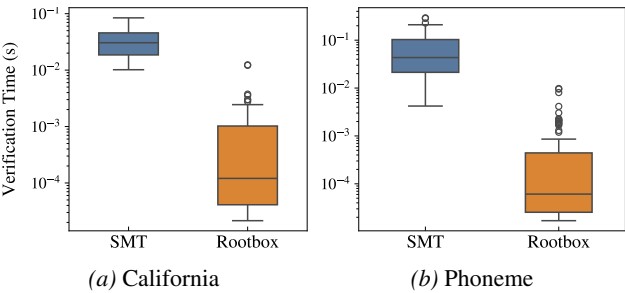

*(a)* California  *(b)* Phoneme

*Figure 5.* Distribution of the average time in seconds over all models on the Pareto front to check if there exists an adversarial example within a certain distance according to a hybrid norm (empirical robustness definition 1) using SMT (Devos et al., 2021a)) and a Rootbox index for the California (left) and Phoneme (right) datasets. The average is computed over 500 correctly classified test examples.

**Average slope** For the Vehicle dataset, we ran a query that finds the minimal value of $c$ for which it holds that for all $x, x' \in \mathcal{X}$, $\frac{|\boldsymbol{T}(x) - \boldsymbol{T}(x')|}{\max(\Delta_x, \delta(x, x'))} \leq c$. The average-slope constraint from Table 1 is then satisfied for any $\Delta_y = c\Delta_x$. The $c$ found here can also be interpreted as a "capped Lipschitz constant", where the cap is imposed by never dividing by anything smaller than $\Delta_x$.

We ran the query on models of varying size, for the $L_\infty$ norm. Runtimes increased roughly linearly with $|\mathcal{O}|$, from 341s ($|\mathcal{O}| = 8.2 \cdot 10^6$) to 1714s ($|\mathcal{O}| = 3.4 \cdot 10^7$). High values for $c$ were found, implying that classification tree ensembles quickly switch from maximally positive to maximally negative near the decision surface (not gradually like some other model classes). For more details, see Appendix E.3.

## 8. Limitations

A core limitation of our approach is that it relies on enumerating and storing all output configurations. This implies that the computational requirements (both time and memory) scale exponentially with the size of the ensemble. Empirically, the number of OCs is often orders of magnitude less

than the upper bounds of Theorems 3.1 and 3.2, and the approach is feasible for a wide range of models (especially when combined with ensemble compression methods, which allow a user to trade some accuracy for verifiability when needed). Nevertheless, there are circumstances where the approach is infeasible, and special-purpose methods with better scaling behavior (such as Kantchelian et al.'s MILP approach) remain preferable.

Our implementation can still be improved in multiple ways. It uses relatively simple index structures, and it is not known how more sophisticated indexes might affect performance. Data structures could be made more memory-efficient: e.g., box borders are currently stored as double-precision floats. The entire index structure is kept in main memory: a more sophisticated implementation could use disk storage. These improvements will typically trade a constant factor in time complexity for wider applicability of the approach.

## 9. Conclusion

We provide a unifying view of verification for tree ensembles by framing tasks as search problems over an ensemble's OC-space (i.e., the set of all possible combinations of individual trees' predictions). This yields a generic and flexible approach that covers a wide variety of existing verification tasks plus some new ones that we propose. Interestingly, verification queries can be answered in time linear or quadratic in the size of the OC-space (which in the worst case is exponential in the ensemble size). The efficiency of the search can be further improved by using index structures that exploit special spatial properties of the OC-space. Empirical results show that fully enumerating the OC-space for reasonably sized learned ensembles is both feasible and not costly (e.g., can be done in less than 15 minutes for many ensembles), and that this approach to verification can handle a wide range of verification tasks while still being able to outperform existing verifiers designed to target a specific verification problem. The one-time cost of enumerating and indexing can be amortized over all subsequent queries, making this approach particularly beneficial for situations when many queries must be answered for the same model.

## Acknowledgements

This research is supported by the Flemish government under the "Onderzoeksprogramma Artificiële Intelligentie (AI) Vlaanderen" programme (TM, LC, HB, JD & WM), Research Foundation Flanders (FWO 11I8125N (LC), FWO G039626N (JD)).

## Impact Statement

This paper presents work whose goal is to advance the field of model verification. It is crucial that models behave as intended and, hence, we need formal guarantees. This helps to detect errors and builds confidence in its reliability, which is especially important in safety-critical or high-stakes applications.

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

# A. Proofs of Theorems

We now provide proofs of the theorems introduced in Sections 3 and 5.

## A.1. Output configurations and OC-space

**Theorem 3.1** (Leaf Bound). $|\mathcal{O}| \leq \prod_m L_m$ with $L_m$ the number of leaves of tree $T_m$.

*Proof of Theorem 3.1.* This directly follows from the fact that output configurations are tuples of leaves, so $\mathcal{O}$ is a subset of the cartesian product of all sets of leaves, which itself has cardinality $\prod_m L_m$. □

**Theorem 3.2** (Feature Bound). $|\mathcal{O}| \leq \prod_f (C_f + 1)$ with $C_f$ the number of constants with which $x_f$ is compared in the ensemble.

*Proof of Theorem 3.2.* Assume $x_f$ is compared to $k$ different constants $c_{f1}, c_{f2}, \ldots, c_{fk_f}$ with $c_{f,j} < c_{f,j+1}$. This partitions the domain of $x_f$ into $k_f + 1$ intervals $I_{f,1} = (-\infty, c_{f1}), I_{f,2} = [c_{f1}, c_{f2}), \ldots, I_{f,k_f+1} = [c_{fk_f}, +\infty)$. Doing this for all $x_f$ partitions $\mathcal{X}$ into $\prod_f (C_f + 1)$ cells of the form $\prod_f I_{f,k_f}$. Now consider any leaf in the ensemble. Its box's $l_f$ and $u_f$ values must coincide with a threshold on $x_f$ used higher up in the tree (or be infinite), and therefore must both be equal to some $c_{f,j}$. As a result, the box must be equal to either a cell or a union of cells. Since boxes cannot overlap, there cannot be more boxes than cells. □

## A.2. Efficiently searching the OC-space

**Theorem 5.1.** *When $T'$ is a subensemble of $T$, the OC-space of $T$ subpartitions the OC-space of $T'$.*

*Proof of Theorem 5.1.* By definition, $T'$ can be obtained by repeatedly performing the following operation on any tree in $T$: take a node just above two leaves and turn this node into a new leaf. We show that applying this operation to an ensemble $T$ with OC-space $\mathcal{O}_T$ results in an OC-space that is subpartitioned by $\mathcal{O}_T$; the theorem then follows by applying transitivity.

Assume the operation is applied to a tree $T_m$, and the leaves being merged are $\lambda_1$ and $\lambda_2$. For all output configurations in $\mathcal{O}_T$, the following holds: either the OC has $\lambda_1$ or $\lambda_2$ as its $m$'th component, or it does not. In the latter case, it is still an output configuration after the operation. In the former case, the OC is replaced by an OC that is equal except it has the new leaf $\lambda$ in place of $\lambda_1$ or $\lambda_2$. Any pair of OC's that are equal except that one has $\lambda_1$ and the other has $\lambda_2$ get mapped to a single OC in this way. Similarly, the box of any output configuration that does not contain $\lambda_1$ or $\lambda_2$ stays

the same, whereas the box of any configuration with $\lambda_1$ (bounded by $l_f \leq x_f < \tau$, with $x_f < \tau$ the test separating $\lambda_1$ and $\lambda_2$) gets merged with the box of the corresponding configuration with $\lambda_2$ ($\tau \leq x_f < u_f$), yielding a single box with borders $l_f \leq x_f < u_f$. The new OC-space thus only contains boxes that were in the original space or are the union of two boxes. This implies that the original OC-space subpartitions the new. □

**Theorem 5.2.** *If the distance from example $x$ to bounding box $B$ exceeds $d$, then the distance from $x$ to any box $b$ that lies entirely inside $B$ must also exceed $d$.*

*Proof of Theorem 5.2.* The distance from $x$ to $B$ is by definition the smallest distance from $x$ to any point $x'$ in $B$, and similar for $b$. Now, since all points $x'$ in $b$ are also contained in $B$, it is impossible that $\min_{x' \in B} \delta(x, x')$ is strictly greater than $\min_{x' \in b} \delta(x, x')$. □

# B. OC-space Bounds

Figure 6 empirically illustrates the relationship between the leaf bound (Theorem 3.1) and feature bound (Theorem 3.2), and the true size of the OC-space for all models on the Pareto front for the Electricity, Volkert, and Adult datasets. The results show that the bounds are complimentary. For the Electricity dataset, the leaf bound clearly provides the best estimates whereas on Volkert the feature bound is tighter. The Adult dataset shows that sometimes neither dominates.

# C. Pseudocode

First, we provide pseudocode for the algorithms in Table 1. Second, we present an algorithm for performing lookups using the indexes introduced in Section 5.

## C.1. Table 1 algorithms

Table 1 concisely described algorithms using aggregation functions over sets. For completeness, Algorithms 1 through 11 provide pseudocode that clearly shows that the algorithms are linear or quadratic in the size of the OC-space. Note that output configurations are stored as labeled boxes $(b, y)$.

## C.2. Pseudocode for indexes

Algorithm 12 shows how an index can be used to efficiently solve the following task: given a box $b$ with label $y$, find the nearest box whose label differs from $y$. We assume a binary classification setting, and we assume positive and negative boxes are stored and indexed separately. The algorithm is equally applicable when an instance $x$ is provided instead of a box $b$, by defining $b$ as a size-0 box with upper and lower bounds for each dimension equal to $x$'s values. In a minor variant, it can be used to efficiently find all boxes

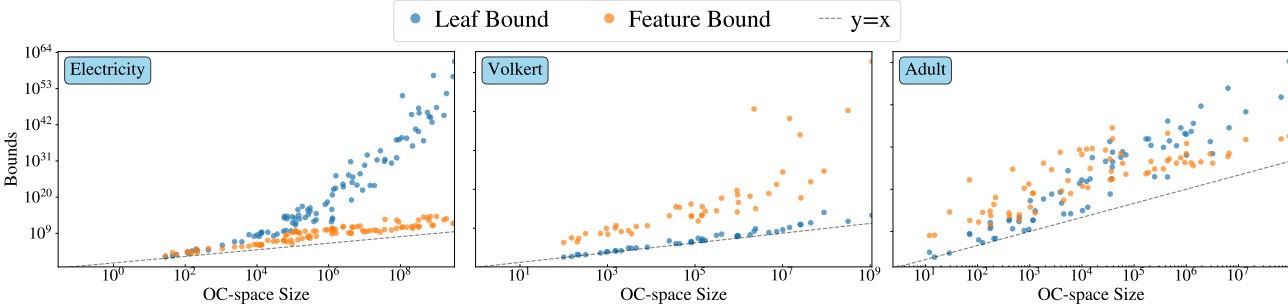

*Figure 6.* The leaf and feature bounds on the size of the OC-space vs. its true size for all models on the Pareto front for the Electricity (left), Volkert (center), and Adult (right) datasets. The dashed line is the identity function and illustrates when the bound would equal the true size of the OC-space. The plots demonstrate the complementarity of the bounds. For Electricity, the feature bound is the tightest, for Volkert the leaf bound is the tightest, and for Adult both provide similar estimates.

---

**Algorithm 1** Plausibility of range

**Input:**
  $\mathcal{O}$: the enumerated OC-space
**Output:**
  *true* if constraint satisfied, else *false*

**for** $(b, y)$ **in** $\mathcal{O}$ **do**
  **if** $y < c_1$ **or** $y > c_2$ **then**
    **return** *false*
  **end if**
**end for**
**return** *true*

---

**Algorithm 2** Empirical robustness 1: No adversarial example within a distance of $\epsilon$

**Input:**
  $\mathcal{O}$: the enumerated OC-space
  $(x, y)$: a labeled instance
  $\epsilon$: the maximum perturbation
**Output:**
  *true* if no adversarial examples within $\epsilon$, else *false*

**for** $b', y'$ **in** $\mathcal{O}$ **do**
  **if** $\text{mindist}(x, b') < \epsilon$ **and** $y \neq y'$ **then**
    **return** *false*
  **end if**
**end for**
**return** *true*

---

**Algorithm 3** Empirical robustness 2: Distance to the closest adversarial example

**Input:**
  $\mathcal{O}$: the enumerated OC-space
  $(x, y)$: a labeled instance
**Output:**
  $\delta$: distance to the closest adversarial example

$\delta \leftarrow \infty$
**for** $(b', y')$ **in** $\mathcal{O}$ **do**
  **if** $\text{mindist}(x, b') < \delta$ **and** $y \neq y'$ **then**
    $\delta \leftarrow \text{mindist}(x, b')$
  **end if**
**end for**
**return** $\delta$

---

$\text{mindist}(b, \ldots) < d^*$ conditions change into $\max(y - m, M - y)/\text{mindist}(b, \ldots) > s^*$ (with $M$ and $m$ the highest and lowest value of $y$ within the bounding box), the $s^*$ update changes similarly, and the initial value for $s^*$ is 0.

## D. Algorithm Correctness Proofs

**Theorem D.1.** *Algorithm 7 checks fairness in* $O(|\mathcal{O}|^2)$ *time.*

*Proof.* The quadratic complexity follows from having two nested for loops. To show correctness, we have to show that (a) a pair $x,x'$ can only violate fairness when $x$ and $x'$ belong to different $p$-dependent boxes $b, b'$ with $\text{relax}(b, p) \cap b' \neq \emptyset$ and $y \neq y'$, and (b) vice versa, whenever $b$ and $b'$ are such boxes, such a pair exists.
(a): A pair $x,x'$ can only violate fairness when $y \neq y'$, which implies $b \neq b'$, and $x_i = x'_i$ for all $i \neq p$. The latter means that $p$ is the only attribute that distinguishes $x$ from $x'$, which in turn implies two things: (1) $b$ and $b'$ are $p$-dependent (otherwise they could not distinguish $x$ from $x'$); (2) since $\text{relax}(b, p)$ is $p$-independent and $x$ belongs to

---

within a given distance $\epsilon$ (the "smallest distance found until now" is then replaced with a fixed distance $\epsilon$, and found boxes are added to a set of solutions), or boxes that may overlap ($\epsilon = 0$). The initial call to the recursive algorithm is SEARCH($b$,$I$,None,$\infty$).

For average-slope constraints, a similar algorithm can be used for finding the greatest average slope; the

**Algorithm 4** Empirical robustness 3: Adversarial example with the largest change in predicted output

> **Input:**
> $\mathcal{O}$: the enumerated OC-space
> $(x, y, b)$: a labeled instance and its OC box $b$
> $\epsilon$: the maximum perturbation
> **Output:**
> $x^*$: an adversarial example that maximizes $|y^* - y|$
>
> $b^*, y^* \leftarrow b, y$
> **for** $(b', y')$ **in** $\mathcal{O}$ **do**
>     **if** $\mathrm{mindist}(x, b') < \epsilon$ **and** $|y - y'| > |y - y^*|$ **then**
>         $b^*, y^* \leftarrow b', y'$
>     **end if**
> **end for**
> // generate an instance $x^*$ in box $b^*$ s.t. $\mathrm{dist}(x, x^*) < \epsilon$.
> $x^* \leftarrow \mathrm{sample}(b^*, x, \epsilon)$
> **return** $x^*$

**Algorithm 5** Average slope

> **Input:**
> $\mathcal{O}$: the enumerated OC-space
> $\Delta_x$: maximal distance considered "nearby"
> $\Delta_y$: allowed $y$ difference between nearby instances
> **Output:**
> *true* if average slope constraint satisfied, else *false*
>
> **for** $(b, y)$ **in** $\mathcal{O}$ **do**
>     **for** $(b', y')$ **in** $\mathcal{O}$ **do**
>         **if** $\mathrm{mindist}(b, b') < \Delta_x$ **and** $|y - y'| > \Delta_y$ **then**
>             **return** *false*
>         **end if**
>     **end for**
> **end for**
> **return** *true*

**Algorithm 6** OC-level robustness

> **Require:** bounded $\mathcal{X}$
> **Input:**
> $\mathcal{O}$: the enumerated OC-space
> $\epsilon$: the maximum perturbation
> **Output:**
> the percentage of $\mathcal{X}$ that is robust
>
> $V = 0$
> **for** $(b, y)$ **in** $\mathcal{O}$ **do**
>     **for** $(b', y')$ **in** $\mathcal{O}$ **do**
>         **if** $\mathrm{mindist}(b, b') < \epsilon$ **and** $y \neq y'$ **then**
>             $V \leftarrow V + \mathrm{vol}(b)$
>             **break**
>         **end if**
>     **end for**
> **end for**
> **return** $1 - \frac{V}{\mathrm{vol}(\mathcal{O})}$

**Algorithm 7** Global fairness

> **Input:**
> $\mathcal{O}$: the enumerated OC-space
> $p$: the protected attribute
> **Output:**
> *true* if model is fair, else *false*
>
> **for** $(b, y)$ **in** $\mathcal{O}$ **do**
>     **for** $(b', y')$ **in** $\mathcal{O}$ **do**
>         **if** $\mathrm{relax}(b, p) \cap b' \neq \emptyset$ **and** $y \neq y'$ **then**
>             **return** *false*
>         **end if**
>     **end for**
> **end for**
> **return** *true*

it, so does $x'$, hence $\mathrm{relax}(b, p) \cap b' \neq \emptyset$.

(b) For any $p$-dependent boxes $b$ and $b'$ with $y' \neq y$ and $\mathrm{relax}(b, p) \cap b' \neq \emptyset$, take an $x'$ in this intersection. Let $x$ be identical to $x'$ except for $x_p$, which is chosen so that $x \in b$. $x$ and $x'$ violate fairness. $\qquad\square$

**Theorem D.2.** *Algorithm 9 checks monotonicity in $x_m$ in time $O(|\mathcal{O}|^2)$.*

*Proof.* This case is similar to fairness, in the sense that the constraint can only be violated by $x, x'$ pairs that are identical on all attributes except $m$. The only difference is that the condition $y \neq y'$ is replaced by $y' < y \wedge u'_m > l_m$. This conjunction is true, together with $\mathrm{relax}(b, m) \cap b' \neq \emptyset$, if and only if there exist $x \in b$ and $x' \in b'$ that are equal except for $x'_m \geq x_m$ while $y' < y$. $\qquad\square$

## E. Detailed Results

**Datasets**   Table 3 summarizes the characteristics of the 16 datasets on which the 1065 ensembles considered in this paper were learned. The multiclass datasets were transformed to binary (i.e., positive versus negative) classification problems in the following ways:

- DryBean: Horoz vs other classes

- FashionMnist: Trouser vs other classes

- Mnist: 2 vs 4

- Volkert: 2 vs 7

**Algorithm 8** Unfair regions

**Input:**
$\mathcal{O}$: the enumerated OC-space
$p$: the protected attribute
**Output:**
unfair_regions: all unfair boxes in $\mathcal{X}$

unfair_regions $\leftarrow \emptyset$
**for** $(b, y)$ **in** $\mathcal{O}$ **do**
  **for** $(b', y')$ **in** $\mathcal{O}$ **do**
    **if** relax$(b, p) \cap b' \neq \emptyset$ **and** $y \neq y'$ **then**
      add relax$(b, p)$ $\cap$ relax$(b', p))$ to unfair_regions
    **end if**
  **end for**
**end for**
**return** unfair_regions

**Algorithm 9** Monotonicity

**Input:**
$\mathcal{O}$: the enumerated OC-space
$m$: the monotonic attribute
**Output:**
*true* if monotonicity satisfied, else *false*

**for** $(b, y)$ **in** $\mathcal{O}$ **do**
  **for** $(b', y')$ **in** $\mathcal{O}$ **do**
    **if** relax$(b, m) \cap b' \neq \emptyset$ **and** $y' > y$ **and** $u'_m \leq l_m$
    **then**
      **return** *false*
    **end if**
  **end for**
**end for**
**return** *true*

**Algorithm 10** Safety

**Input:**
$\mathcal{O}$: the enumerated OC-space
$\mathcal{O}_{ref}$: a reference enumerated OC-space
**Output:**
max_deviation: $\max_{\mathcal{X}} |y' - y|$

max_deviation $\leftarrow 0$
**for** $(b, y)$ **in** $\mathcal{O}$ **do**
  **for** $(b', y')$ **in** $\mathcal{O}_{ref}$ **do**
    **if** $b \cap b' \neq \emptyset$ **and** $|y' - y| >$ result **then**
      max_deviation $\leftarrow |y' - y|$
    **end if**
  **end for**
**end for**
**return** max_deviation

**Algorithm 11** Multiplicity

**Require:** bounded $\mathcal{X}$
**Input:**
$\mathcal{O}$: the enumerated OC-space
$\mathcal{O}_{ref}$: a reference enumerated OC-space
$\epsilon$: the maximum perturbation
**Output:**
the percentage of $\mathcal{X}$ s.t. $|y' - y| > \epsilon$

variable_regions $\leftarrow \emptyset$
**for** $(b, y)$ **in** $\mathcal{O}$ **do**
  **for** $(b', y')$ **in** $\mathcal{O}_{ref}$ **do**
    **if** $b \cap b' \neq \emptyset$ **and** $|y - y'| > \epsilon$ **then**
      add $(b \cap b')$ to variable_regions
    **end if**
  **end for**
**end for**
**return** $vol$(variable_regions)$/vol(\mathcal{X})$

**Algorithm 12** Search

**Input:**
$b$: given box
$I$: tree-structured index
$b^*$: nearest adversarial box found until now
$d^*$: distance from $b$ to $b^*$
**Output:**
$(b^*, d^*)$: nearest box and distance to it

**if** $I$ is a leaf **then**
  **for all** $b'$ stored in $I$ **do**
    **if** mindist$(b, b') < d^*$ **then**
      $d^* \leftarrow$ mindist$(b, b')$
      $b^* \leftarrow b'$
    **end if**
  **end for**
**else**
  **for all** children $C$ of $I$ **do**
    **if** mindist$(b,$ bounding_box$(C)) < d^*$ **then**
      $(b^*, d^*) \leftarrow$ Search$(b, C, b^*, d^*)$
    **end if**
  **end for**
**end if**
**return** $(b^*, d^*)$

### E.1. Enumerating and indexing the OC-space

Table 4 shows for which datasets we could enumerate all models on the pareto front. We find that if $|\mathcal{O}| < 10^{10}$ the full OC-space can typically be enumerated within 24h. If one has more time, it is possible to enumerate larger models. However, from the fact that all models were enumerated for 9/16 datasets, we find that in practice compressed models

*Table 3.* Properties of the datasets: Name, number of examples, number of features, and class prior $\alpha$ (i.e., proportion of positive examples).

| Name | # samples | # features | $\alpha$ |
|---|---|---|---|
| Adult | 48842 | 32 | 0.24 |
| California | 20634 | 8 | 0.50 |
| Compas | 4966 | 11 | 0.50 |
| Credit | 16714 | 10 | 0.50 |
| DryBean | 13611 | 16 | 0.19 |
| Electricity | 38474 | 8 | 0.50 |
| FashionMnist | 70000 | 784 | 0.10 |
| Higgs | 940160 | 24 | 0.50 |
| Ijcnn | 141691 | 22 | 0.10 |
| Jannis | 57580 | 54 | 0.50 |
| MiniBooNE | 72998 | 50 | 0.50 |
| Mnist | 13814 | 784 | 0.49 |
| Phoneme | 5404 | 5 | 0.71 |
| Spambase | 4601 | 57 | 0.39 |
| Vehicle | 846 | 18 | 0.49 |
| Volkert | 24325 | 180 | 0.53 |

*Table 4.* For each dataset, the number of enumerated models, total models, the percentage of enumerated models, and the size of the largest enumerated OC-space when considering all models on the Pareto front. The considered models were obtained using LOP.

| Dataset | Enumerated | Total | Percentage | Max $|\mathcal{O}|$ |
|---|---|---|---|---|
| Adult | 70 | 70 | 100.0 | 9.45e+07 |
| California | 70 | 70 | 100.0 | 2.59e+09 |
| Compas | 55 | 55 | 100.0 | 2.48e+02 |
| Credit | 60 | 60 | 100.0 | 3.75e+05 |
| DryBean | 35 | 35 | 100.0 | 1.56e+03 |
| Electricity | 98 | 115 | 85.2 | 3.23e+09 |
| FashionMnist | 53 | 55 | 96.4 | 3.41e+09 |
| Higgs | 55 | 105 | 52.4 | 1.47e+10 |
| Ijcnn | 75 | 125 | 60.0 | 1.24e+10 |
| Jannis | 38 | 60 | 63.3 | 5.81e+09 |
| MiniBooNE | 39 | 80 | 48.8 | 1.55e+10 |
| Mnist | 39 | 40 | 97.5 | 1.85e+09 |
| Phoneme | 85 | 85 | 100.0 | 8.14e+07 |
| Spambase | 30 | 30 | 100.0 | 4.97e+07 |
| Vehicle | 35 | 35 | 100.0 | 2.52e+09 |
| Volkert | 45 | 45 | 100.0 | 1.10e+09 |
| Total | 882 | 1065 | 82.8 | - |

are often smaller than this.

Table 5 shows the characteristics of the model with the largest OC-space size that was enumerated for each dataset. The number of trees varies widely, with the largest models typically containing between 15-45 trees and at most 67 (Electricity). The maximum tree depth in an ensemble is typically four; the highest value observed is eight (Phoneme). There are models that could be enumerated with more or deeper trees, but they had a smaller OC-space size. For instance, on the Electricity dataset there are multiple models on the Pareto front that have trees with a maximum depth of eight that were enumerated. It is important to interpret these results in context: these are the highest values observed for models *on the Pareto front*. Systematically varying model sizes would give us more insight, but how to do this is not obvious given the complex interplay among the trees in an ensemble and their effect on the size of the OC-space (e.g., see Section 3 and Appendix B).

Figure 7 shows the time required to build Rootbox and OC-Tree indexes for each dataset. Building an index on top of an enumerated OC-space is often faster than enumerating the models. Both Rootbox and OC-Tree indexes are typically built within 15 minutes.

### E.2. Comparison against special-purpose approaches

**Adversarial robustness**  Table 6 reports the geometric mean of the models' average verification times in milliseconds for each dataset where all models on the Pareto front were enumerated. On average, Kantchelian et al.'s approach is a factor of 10 slower than searching the OC-space (i.e.,

*Table 5.* Characteristics of the largest enumerated model in terms of OC-space size on each dataset: M denotes the number of trees, D is the maximum depth of a tree and $|\mathcal{O}|$ is the size of its OC-space.

| Dataset | M | D | #Leaves | Max $|\mathcal{O}|$ |
|---|---|---|---|---|
| Adult | 27 | 4 | 109 | 9.45e+07 |
| California | 39 | 4 | 489 | 2.59e+09 |
| Compas | 6 | 4 | 30 | 2.48e+02 |
| Credit | 17 | 4 | 75 | 3.75e+05 |
| DryBean | 6 | 4 | 33 | 1.56e+03 |
| Electricity | 67 | 4 | 612 | 3.23e+09 |
| FashionMnist | 9 | 6 | 155 | 3.41e+09 |
| Higgs | 39 | 4 | 194 | 1.47e+10 |
| Ijcnn | 23 | 4 | 238 | 1.24e+10 |
| Jannis | 20 | 4 | 127 | 5.81e+09 |
| MiniBooNE | 16 | 4 | 124 | 1.55e+10 |
| Mnist | 12 | 4 | 101 | 1.85e+09 |
| Phoneme | 44 | 8 | 1135 | 8.14e+07 |
| Spambase | 16 | 4 | 80 | 4.97e+07 |
| Vehicle | 37 | 6 | 171 | 2.52e+09 |
| Volkert | 7 | 6 | 239 | 1.10e+09 |

for any indexing). Figure 8 shows that a linear scan over a list of positives or negatives (i.e., depending on the target label of the attack) scales worse than the other approaches. How the other three approaches scale varies according to the dataset. Regardless of the size of the OC-space, using either a rootbox or OC-tree index outperforms Kantchelian et al.'s approach for the vast majority of the cases. Which

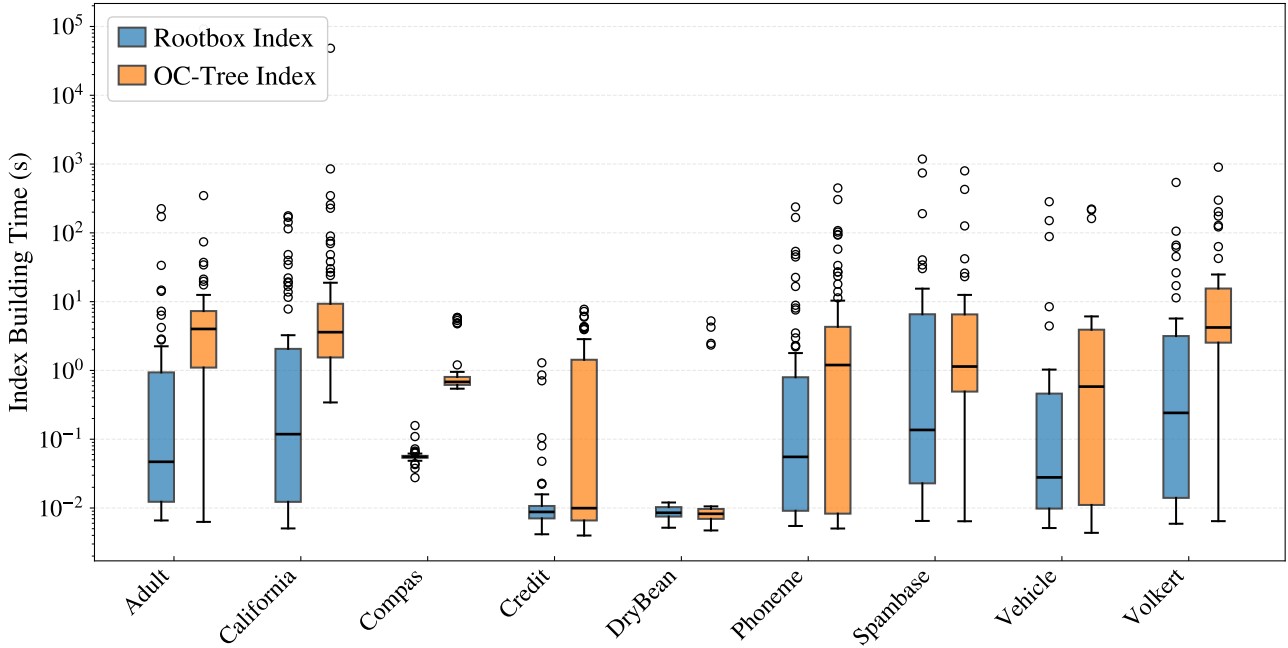

*Figure 7.* The distribution of time in seconds needed to build a Rootbox index and OC-Tree index for all models on the Pareto front for each dataset where all models could be enumerated within 24 hours.

of the rootbox or OC-tree index is better varies by dataset.

*Table 6.* Geometric mean time (in milliseconds) required to find the closest adversarial example to an instance using Kantchelian et al.'s MILP approach, a Linear Scan of the OC-space, a Rootbox index, and an OC-Tree index, for each dataset where all models could be enumerated within 24 hours. For each dataset, the geometric mean is computed by first averaging the verification time over 500 correctly classified instances for each model on the Pareto front, and then taking the geometric mean across all models.

| Dataset | Kantchelian | Linear Scan | Rootbox | OC-Tree |
|---|---|---|---|---|
| Adult | 7.153 | 0.696 | 0.390 | 1.515 |
| California | 12.407 | 2.376 | 0.208 | 1.687 |
| Compas | 1.932 | 0.008 | 0.365 | 1.208 |
| Credit | 3.418 | 0.018 | 0.056 | 0.027 |
| DryBean | 3.559 | 0.014 | 0.052 | 0.008 |
| Phoneme | 24.504 | 0.482 | 0.265 | 0.467 |
| Spambase | 5.545 | 1.327 | 0.326 | 0.493 |
| Vehicle | 6.287 | 0.366 | 0.186 | 0.170 |
| Volkert | 8.319 | 4.885 | 0.697 | 1.927 |

Table 2 shows the number of queries after which the cumulative run time of Kantchelian et al.'s approach is greater than the cumulative run time for using a Rootbox index or an OC-Tree index (i.e., including the time needed to enumerate and index the OC-space). This is computed by dividing the upfront cost (enumeration plus index construction time) by the average per-query speedup of the indexed search over Kantchelian et al.'s approach, measured over 500 correctly classified test instances per model. The resulting quantity is then averaged over all Pareto front models for which the index-based approach is faster than Kantchelian et al.'s approach to answer a closest adversarial example query. The 3.1% (12.1%) of models for which Kantchelian et al. was faster than the Rootbox (OC-Tree) index search are excluded, as amortization is not defined in those cases.

**Fairness** Table 7 shows detailed results for verifying lack of causal discrimination (fairness) on the Adult dataset. For one model (M=50, D=4, LR=0.25, fold 2) the OC-Tree search timed out before it could enumerate all unfair boxes.

Table 8 shows the detailed results on the Compas dataset. We were able to verify all 55 models on the Pareto front. However, as is clear from Table 4, most of these models are very simple to verify, having at most 248 OCs.

### E.3. Flexibility in the verifiable properties

**Average slope** Table 9 shows that even though the query is in principle quadratic in $|\mathcal{O}|$, in practice it behaves more or less linearly. The larger OC-space subdivides $\mathcal{X}$ into smaller boxes but this does not affect the obtained values for $c$, $\delta(x, x')$ or $|y - y'|$ much. The range $|y - y'|$ is always close to the full range of the ensemble (e.g., for the 8.2M model, the highest and lowest predictions found are 9.6 and $-10.2$ so the full range is 19.8), which means that $T$ very quickly changes from one extreme to the other near the decision border.

*Table 7.* Detailed results for fairness verification on the Adult dataset. The first column indicates the hyperparameters used to learn the uncompressed model (M: #Trees, D: Max depth, LR: Learning rate). We report the number of boxes where the model is unfair and verification time (s) of Veritas and OC-Tree search.

| M | D | LR | Fold | #Boxes | Veritas (s) | OC-Tree (s) |
|---|---|----|------|--------|-------------|-------------|
| 10 | 4 | 0.5 | 0 | 11 | 0.640 | 0.158 |
|    |   |     | 1 | 25 | 0.767 | 0.139 |
|    |   |     | 2 | 35 | 0.004 | 0.141 |
|    |   |     | 3 | 395 | 0.111 | 0.148 |
|    |   |     | 4 | 30 | 0.036 | 0.146 |
| 10 | 4 | 1.0 | 0 | 646 | 0.429 | 0.328 |
|    |   |     | 1 | 18977 | 11.581 | 3.599 |
|    |   |     | 2 | 303 | 0.211 | 0.190 |
|    |   |     | 3 | 20 | 0.008 | 0.146 |
|    |   |     | 4 | 553 | 0.121 | 0.155 |
| 10 | 6 | 0.1 | 0 | 0 | 0.004 | 0.000 |
|    |   |     | 1 | 2 | 0.002 | 0.122 |
|    |   |     | 2 | 0 | 0.015 | 0.001 |
|    |   |     | 3 | 0 | 0.003 | 0.001 |
|    |   |     | 4 | 0 | 0.003 | 0.000 |
| 25 | 4 | 0.1 | 0 | 12 | 0.003 | 0.145 |
|    |   |     | 1 | 101 | 0.027 | 0.133 |
|    |   |     | 2 | 12 | 0.002 | 0.151 |
|    |   |     | 3 | 44 | 0.012 | 0.147 |
|    |   |     | 4 | 38 | 0.006 | 0.138 |
| 25 | 4 | 0.5 | 0 | 88176 | 311.275 | 11.987 |
|    |   |     | 1 | 668525 | 5070.678 | 1010.595 |
|    |   |     | 2 | 77584 | 1350.257 | 63.337 |
|    |   |     | 3 | 643897 | 3383.883 | 432.733 |
|    |   |     | 4 | 82579 | 52.071 | 19.578 |
| 25 | 8 | 0.25 | 0 | 0 | 0.003 | 0.000 |
|    |   |     | 1 | 141 | 0.040 | 0.130 |
|    |   |     | 2 | 37 | 0.190 | 0.239 |
|    |   |     | 3 | 170 | 0.642 | 0.144 |
|    |   |     | 4 | 1119 | 0.576 | 0.169 |
| 25 | 8 | 0.5 | 0 | 296 | 0.089 | 0.241 |
|    |   |     | 1 | 16573 | 2.561 | 2.419 |
|    |   |     | 2 | 15 | 0.003 | 0.106 |
|    |   |     | 3 | 1468 | 1.345 | 0.333 |
|    |   |     | 4 | 18860 | 114.953 | 2.032 |
| 50 | 4 | 0.25 | 0 | 8916 | 8.844 | 3.012 |
|    |   |     | 1 | 2557 | 1.249 | 0.352 |
|    |   |     | 2 | 2415852 | 75822.033 | TIME_OUT |
|    |   |     | 3 | 31979 | 44.236 | 33.254 |
|    |   |     | 4 | 69721 | 101.461 | 324.883 |
| 50 | 4 | 1.0 | 0 | 117048 | 3046.773 | 79.245 |
|    |   |     | 1 | 1008 | 1.471 | 1.217 |
|    |   |     | 2 | 193 | 0.137 | 0.773 |
|    |   |     | 3 | 20 | 0.009 | 0.168 |
|    |   |     | 4 | 38671 | 4.818 | 7.196 |
| 50 | 8 | 0.5 | 0 | 673 | 0.390 | 0.220 |
|    |   |     | 1 | 960 | 0.535 | 0.295 |
|    |   |     | 2 | 29 | 0.081 | 0.114 |
|    |   |     | 3 | 0 | 0.008 | 0.005 |
|    |   |     | 4 | 361 | 2.111 | 0.135 |
| 100 | 4 | 0.1 | 0 | 47764 | 97.458 | 354.416 |
|     |   |     | 1 | 14505 | 9.989 | 40.459 |
|     |   |     | 2 | 2671 | 2.917 | 0.658 |
|     |   |     | 3 | 2819 | 0.806 | 1.007 |
|     |   |     | 4 | 96086 | 15.482 | 163.698 |
| 100 | 4 | 0.25 | 0 | 35154 | 45.825 | 77.699 |
|     |   |     | 1 | 5056 | 5.020 | 1.059 |
|     |   |     | 2 | 1892830 | 1135.000 | 64169.850 |
|     |   |     | 3 | 29978 | 9.265 | 28.908 |
|     |   |     | 4 | 199919 | 243.460 | 13608.936 |
| 100 | 6 | 0.1 | 0 | 2771 | 5.127 | 1.101 |
|     |   |     | 1 | 1888 | 0.606 | 0.564 |
|     |   |     | 2 | 1123 | 1.328 | 0.541 |
|     |   |     | 3 | 60985 | 32.862 | 18.386 |
|     |   |     | 4 | 4840 | 6.783 | 1.637 |
| 100 | 6 | 1.0 | 0 | 60 | 0.010 | 0.333 |
|     |   |     | 1 | 67 | 0.010 | 0.386 |
|     |   |     | 2 | 9 | 0.004 | 0.121 |
|     |   |     | 3 | 43 | 0.003 | 0.113 |
|     |   |     | 4 | 19 | 0.003 | 0.109 |

*Table 8.* Detailed results for fairness verification on the Compas dataset. The first column indicates the hyperparameters used to learn the uncompressed model (M: #Trees, D: Max depth, LR: Learning rate). We report the number of boxes where the model is unfair and verification time (s) of Veritas and OC-Tree search.

| M | D | LR | Fold | #Boxes | Veritas (s) | OC-Tree (s) |
|---|---|----|------|--------|-------------|-------------|
| 25 | 4 | 0.25 | 0 | 0 | 3.84e-03 | 4.13e-04 |
|    |   |     | 1 | 2 | 2.03e-03 | 1.62e-01 |
|    |   |     | 2 | 6 | 2.13e-03 | 1.48e-01 |
|    |   |     | 3 | 3 | 2.07e-03 | 1.48e-01 |
|    |   |     | 4 | 0 | 2.00e-03 | 3.31e-04 |
| 25 | 4 | 0.5 | 0 | 0 | 1.94e-03 | 2.41e-04 |
|    |   |     | 1 | 0 | 2.04e-03 | 1.16e-04 |
|    |   |     | 2 | 0 | 2.02e-03 | 3.81e-04 |
|    |   |     | 3 | 1 | 2.03e-03 | 1.04e-01 |
|    |   |     | 4 | 1 | 2.08e-03 | 1.17e-01 |
| 25 | 4 | 1.0 | 0 | 0 | 2.03e-03 | 9.99e-05 |
|    |   |     | 1 | 0 | 2.02e-03 | 1.09e-04 |
|    |   |     | 2 | 0 | 1.99e-03 | 1.15e-04 |
|    |   |     | 3 | 0 | 2.35e-03 | 1.48e-04 |
|    |   |     | 4 | 1 | 2.06e-03 | 1.45e-01 |
| 25 | 6 | 0.25 | 0 | 0 | 1.97e-03 | 9.94e-05 |
|    |   |     | 1 | 0 | 2.00e-03 | 1.07e-04 |
|    |   |     | 2 | 0 | 2.18e-03 | 3.59e-04 |
|    |   |     | 3 | 1 | 2.06e-03 | 1.04e-01 |
|    |   |     | 4 | 1 | 2.04e-03 | 9.58e-02 |
| 25 | 8 | 0.1 | 0 | 0 | 2.15e-03 | 1.53e-04 |
|    |   |     | 1 | 0 | 2.13e-03 | 2.97e-04 |
|    |   |     | 2 | 0 | 1.98e-03 | 1.05e-04 |
|    |   |     | 3 | 0 | 2.01e-03 | 9.54e-02 |
|    |   |     | 4 | 0 | 2.00e-03 | 1.06e-04 |
| 50 | 4 | 1.0 | 0 | 0 | 1.99e-03 | 1.43e-04 |
|    |   |     | 1 | 0 | 2.00e-03 | 1.03e-04 |
|    |   |     | 2 | 0 | 2.00e-03 | 1.07e-04 |
|    |   |     | 3 | 0 | 2.12e-03 | 1.07e-04 |
|    |   |     | 4 | 0 | 2.00e-03 | 1.06e-04 |
| 50 | 8 | 1.0 | 0 | 0 | 2.14e-03 | 1.51e-04 |
|    |   |     | 1 | 0 | 2.23e-03 | 1.04e-04 |
|    |   |     | 2 | 0 | 2.02e-03 | 1.07e-04 |
|    |   |     | 3 | 0 | 2.00e-03 | 1.09e-04 |
|    |   |     | 4 | 0 | 2.17e-03 | 1.11e-04 |
| 100 | 4 | 0.5 | 0 | 0 | 2.03e-03 | 1.08e-04 |
|     |   |     | 1 | 0 | 2.07e-03 | 3.09e-04 |
|     |   |     | 2 | 0 | 2.05e-03 | 1.11e-04 |
|     |   |     | 3 | 12 | 4.27e-03 | 1.41e-01 |
|     |   |     | 4 | 0 | 2.00e-03 | 1.06e-04 |
| 100 | 6 | 0.25 | 0 | 0 | 2.01e-03 | 1.06e-04 |
|     |   |     | 1 | 0 | 2.01e-03 | 3.03e-04 |
|     |   |     | 2 | 0 | 1.99e-03 | 1.08e-04 |
|     |   |     | 3 | 0 | 2.01e-03 | 1.01e-04 |
|     |   |     | 4 | 1 | 2.32e-03 | 1.04e-01 |
| 100 | 6 | 0.5 | 0 | 0 | 1.93e-03 | 1.05e-04 |
|     |   |     | 1 | 0 | 2.08e-03 | 1.08e-04 |
|     |   |     | 2 | 0 | 2.05e-03 | 9.92e-05 |
|     |   |     | 3 | 0 | 2.07e-03 | 3.36e-04 |
|     |   |     | 4 | 0 | 1.95e-03 | 1.10e-04 |
| 100 | 8 | 1.0 | 0 | 0 | 2.43e-03 | 1.02e-04 |
|     |   |     | 1 | 0 | 2.01e-03 | 1.68e-04 |
|     |   |     | 2 | 0 | 1.99e-03 | 1.01e-04 |
|     |   |     | 3 | 0 | 2.14e-03 | 1.06e-04 |
|     |   |     | 4 | 0 | 2.02e-03 | 7.61e-04 |

*Table 9.* Results for determining the highest average slope over a distance of at least $\Delta_x$ on the Vehicle dataset. $|y - y'|$ indicates the difference in output value between two instances, $\delta(x, x')$ is the distance between the closest pair of instances for which this difference was found. $c$ is the capped Lipschitz constant (which is $|y - y'|/\max(\Delta_x, \delta(x, x')))$.

| $|\mathcal{O}|$ | Time (s) | $\Delta_x$ | $\delta(x, x')$ | $|y - y'|$ | $c$ |
|---|---|---|---|---|---|
| 8.2M | 399 | 1 | 0.200 | 19.76 | 19.76 |
| | 341 | 0.1 | 0.080 | 19.21 | 192.10 |
| | 346 | 0.01 | 0.000 | 17.59 | 1759.00 |
| 12.7M | 632 | 1 | 0.200 | 20.78 | 20.78 |
| | 547 | 0.1 | 0.083 | 20.23 | 202.30 |
| | 543 | 0.01 | 0.000 | 18.61 | 1861.00 |
| 34M | 1714 | 1 | 0.220 | 21.67 | 21.67 |
| | 1494 | 0.1 | 0.083 | 20.55 | 205.50 |
| | 1442 | 0.01 | 0.000 | 19.14 | 1914.00 |

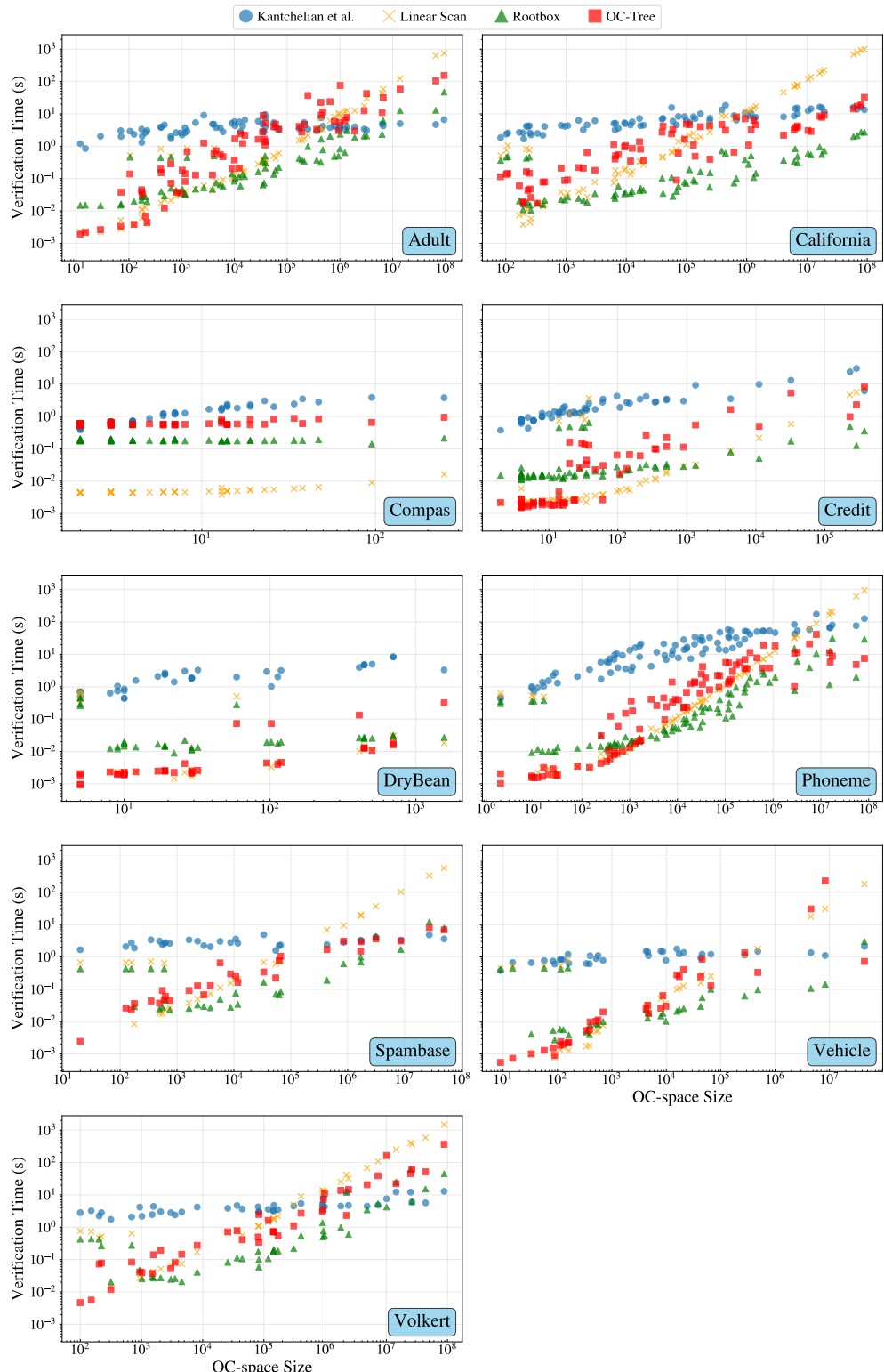

*Figure 8.* The average time in seconds to find the nearest adversarial example (empirical robustness definition 2) as a function of the size of the OC-space using Kantchelian et al.'s MILP approach, a Linear Scan of the OC-space, a Rootbox index, and an OC-Tree index for all models on the Pareto front for each dataset where all models could be enumerated within 24 hours. The average is computed over 500 correctly classified test examples.

