# OpenReview forum: "OC-space: a Unifying Perspective on Verification of Tree Ensembles"
_ICML.cc/2026/Conference — ICML 2026 spotlight_

### Official Review · Reviewer_17JS · 2026-02-21

**Soundness:** 2
**Presentation:** 3
**Significance:** 2
**Originality:** 2
**Overall Recommendation:** 4
**Confidence:** 4

**Summary:**

This paper proposes an efficient verification of tree ensembles using bounding boxes of the OC-space as indices, where the OC-space is the set of all possible combinations of individual trees' predictions. The authors experimentally show the superiority of their proposed method in verification time using XGBoost and 14 benchmark datasets.

**Compliance With Llm Reviewing Policy:**

Affirmed.

**Final Justification:**

I changed my score to weak accept.
Appropriateness I questioned is optimality. There are many index candidates, and I would like to know how good the proposed indices among them. In there experiments, effectiveness of their proposed indices are shown. So, I understood that they are not bad,
but the question is still unclear.

**Key Questions For Authors:**

[Q1] Would you let me know how to use the indices to speed up the algorithms shown in Table 1?
  Can all the algorithms be sped up?

[Q2] What are the advantages and disadvantages of each proposed index?
When should the rootbox index be used, and when should the OC-Tree index be used?

**Limitations:**

The limitations of the proposed indices should be discussed.

**Strengths And Weaknesses:**

[strengths]

・Experimental results demonstrate the superiority of the proposed method in verification time.

・The manuscript is well structured.

・The proposed efficient verification of tree ensembles is practically useful.

・The proposed rootbox and OC-Tree indices look new.

[weaknesses]

・No theoretical support is provided for the superiority of the proposed method in verification time.

・How to use the indices to speed up the algorithms shown in Table 1 is unclear.

・The idea of the proposed rootbox and OC-Tree indices seems easy and not so novel.

---

> ### Author Rebuttal · Authors · 2026-03-31
>
> We would like to thank the reviewer for their reviewing efforts. Below we answer their remaining questions and respond to the main concerns.
>
> **“The idea for the indices seems easy and not so novel”**
>
> This slightly contradicts with the positive point that they are new. Our main contribution is the realization that once one defines OC-space as a storable and searchable object, it is possible for an index structure to be used to answer many types of queries efficiently. We have investigated what spatial indices are available in a couple of databases and open-source libraries (SQLite, PostgreSQL, MLPack's XTree), and these tended to be tailored to much lower dimensions or limited to indexing points and not boxes in the input space. E.g., sqlite only offers spatial indices for maximally 5 dimensions.
>
> **"No theoretical support is provided for the superiority of the proposed method in verification time."**
>
> Note that we are discussing NP-hard problems so all discussed approaches have the same worst case behavior of being exponential in ensemble size. While the OC space is exponential in ensemble size, the paper shows that queries can be answered in polynomial time in the size of the OC space. In the best case tree-structured spatial indices yield log time lookup (due to backtracking, the worst case is linear).
>
> **Q1: Would you let me know how to use the indices to speed up the algorithms shown in Table 1? Can all the algorithms be sped up?**
>
> Spatial indexes store objects in a tree structure where each node is annotated with a bounding box. When searching the index, if the distance from a query instance x (or box b) to a bounding box B exceeds c, then the distance from x to any object in B must also exceed c, and therefore the subtree associated with B can be skipped in its entirety.  This makes enumerating all boxes within distance c of a given x efficient. Setting c=0 generates only boxes touching or containing x (or, for a box b: touching or overlapping with b). Our index additionally stores a bounding box for the target values y, which is used for the average slope constraint (see section 4.2 for details) and indexes positive and negative boxes separately, which speeds up queries involving (b,b') pairs with a y != y' condition on them (b is chosen from the positives and b' from the negatives).
>
> **Q2: What are the advantages and disadvantages of each proposed index? When should the rootbox index be used, and when should the OC-Tree index be used?**
>
> We do not have a very clear view on the (dis)advantages of the Rootbox vs. OC-Tree index. Neither consistently outperforms the other, but both clearly improve over linear scans. We did not optimize the indexes for performance: e.g., the OC-Tree is a hierarchical index whose depth can be varied; we consistently chose max(10, M-2) as depth (with M the ensemble size), but better choices might make it more efficient.  More work could certainly be done on optimizing the indexes and studying their performance in detail, but that is beyond the scope of this paper: a detailed comparison between these two specific index structures has limited value as the index structures themselves are somewhat arbitrary.

---

> > ### Author Rebuttal · Reviewer_17JS · 2026-04-04
> >
> > Thank you for your response.
> > As for index structures, I am not sure that the proposed indices are appropriate for the considered verification tasks.

---

> > > ### Author Response · Authors · 2026-04-08
> > >
> > > We are not sure what the reviewer is referring to when questioning the appropriateness of the proposed indices.
> > >
> > > If referring to **speed-up**, Figure 3 in the main text and Figure 7 in the appendix indicate better scaling when using either of the two indexes compared to a linear scan over all the OCs. Additionally, verifying fairness without an index (i.e., comparing all pairs of OCs to find all violations of fairness as in Theorem B.1) quickly becomes infeasible. For a model with |OC|=650k on the Adult dataset, it took 7h to verify fairness without an index compared to 6s with an index and 26s using Veritas.
> > >
> > > If referring to **correctness**: empirically, our method gave the same solutions as other methods, regardless of whether indexes were used. This is a strong indication that the implementation is correct.  Correctness of the method can be shown more formally as follows (we can include proofs in the appendix if that is desired).
> > >
> > > First, recall that all our indexes have a tree structure.  All boxes are stored in the leaves, and with each internal node is associated a "bounding box” with the property that all boxes stored in leaves underneath this node fit entirely inside the bounding box.
> > >
> > > We now have the following theorem:
> > >
> > > **Theorem:** If the distance from example x to bounding box B exceeds d, then the distance from x to any box b that lies entirely inside B must also exceed d.
> > >
> > > Proof:
> > >
> > > The distance from x to B is by definition the smallest distance from x to any point x’ in B, and similar for b.  Now, since all points x’ in b are also contained in B, it is impossible that $min_{x’ \in B} d(x,x’)$ is strictly greater than $min_{x’ \in b} d(x,x’)$. Q.E.D.
> > >
> > > Our indexes utilize this property to speed up verification. When searching for all boxes within a given distance d*, we avoid enumerating boxes inside any bounding box B whose distance to x exceeds d*. The only way in which this could yield incorrect results (i.e. a result different from what a complete enumeration would find) is if it somehow prunes away the true solution. Since the above theorem has been proven, this cannot happen, and the correctness of our index-based search is proven.
> > >
> > > We can apply this theorem in various ways to speed up the algorithms in Table 1. Below we give two examples of properties discussed in the paper, but it can be applied to any property in Table 1.
> > >
> > > **Example 1:** The case where we look for the nearest box, rather than all boxes within a given distance d* of instance x (i.e., adversarial robustness in Section 7.2), is treated by constantly updating d* to the smallest distance found until now. Consequently, as d* decreases, increasingly more bounding boxes can be pruned away.
> > >
> > > **Example 2:** For the average slope constraint (Section 7.3), we rely again on the same principle, but it also takes the range of y into account. The average slope from an example (x, y) to any (x’,y') in B cannot exceed max(y-m,M-y) / d(x,B) with m and M respectively the smallest and largest value of y’ found in B and d(x,B) the smallest distance from x to any point x’ in B. That is because |y’-y| cannot exceed the numerator and d(x,x’) cannot be smaller than d(x,B). Hence, similarly to example 1, all boxes in a bounding box B with max(y-m,M-y) / d(x,B) <= c (with c the greatest slope found till now) can be pruned away when looking for the smallest global upper bound on the slope.

---

### Official Review · Reviewer_Nd9T · 2026-03-13

**Soundness:** 2
**Presentation:** 4
**Significance:** 2
**Originality:** 3
**Overall Recommendation:** 3
**Confidence:** 3

**Summary:**

This paper presents a unifying lens to view certain verification/search problems on tree ensembles. Canonical examples include deciding if there exists an input with $\epsilon$ from a given input $x$ with a different label, or whether the classifier satisfies certain fairness properties such as demographic parity. The authors encourage thinking about these problems by OC-space, which can be thought of as partition of the input space based on which tuple of leaf nodes they get assigned too. They argue that many verification queries are "easy" (=poly in size of OC-space), and supplement their work with experiments on real-world datasets.

**Compliance With Llm Reviewing Policy:**

Affirmed.

**Final Justification:**

Interesting paper overall, but I believe the authors are overclaiming on the novelty front. The authors have added many additional experiments (including in response to other reviewers) that I think strengthen the empirics in the paper. I believe the paper could be vastly improved before publication (better contextualization in related existing work, better demonstration of real world utility etc), but I would not be opposed to acceptance if there are other strong advocates.

**Key Questions For Authors:**

1) How were the 14 datasets chosen for the experiments? Most of them seem like small, fairly dated, datasets, which don't help fully evaluate the practicality of the approach. For instance, why not evaluate on slightly bigger ones (e.g. Higgs or Fashion MNIST which were used in Chen et al. 2019)?

2) For the comparisons in Section 7.2, how were the baselines chosen and implemented? If you used off-the-shelf solvers such as Gurobi, who were the configurations chosen?

3) While I understand that this question is perhaps a little against the spirit of "verification", how do OC-space based methods perform when placed next to methods that allow for approximation? For example, for the problem of finding close adversarial examples, how does this compare to Veritas or the coordinate descent based method in Kantchelian et al. 2016?

**Limitations:**

yes

**Strengths And Weaknesses:**

Strengths: The core idea of the paper is interesting: if the OC-space can be computed efficiently, then it is exciting and helpful to think of a whole suite of verification problems as being answered by "one tool".

Weaknesses: The experimental results are weak and leave much to be desired. The practicality and usefulness of the approach hinges heavily on the whether the OC-space (which can have exponential size) can be efficiently enumerated, and this has not been sufficiently addressed. Also from a novelty perspective, many prior works (as acknowledged in the paper) have considered this notion of OC-space and related framings of tuples of decision tree leaves; the novelty is in trying to use this as a method to solve verification problems that practitioners care about. However, this only works if it is possible to efficiently enumerate in practice in the first place. Further, from a technical perspective, the algorithms to go from OC -> verification application are fairly straightforward; if the possibility of a unifying perspective that is novel. Lastly, the comparisons to existing methods seem far from comprehensive and experiments feel a little cherry-picked.

---

> ### Author Rebuttal · Authors · 2026-03-31
>
> Thank you for the feedback.
>
> **The approach hinges on whether the OC-space can be enumerated**
> Correct. But, we show that this is feasible in more cases than one might expect. On 9 of 14 datasets all models could be enumerated; only on 2 datasets could less than 60% be enumerated. Compression algorithms (e.g., LOP) let users trade off accuracy vs. enumerability. The LOP paper reports significant compression at 0.5% accuracy loss (also the maximal loss we allowed). This can be changed based on user needs.
>
> An alternative view: practitioners may want to tradeoff performance for enumerability so that verification becomes feasible. Interpretability uses the same motivation: sacrificing (a small amount of) accuracy to yield more interpretable models can be warranted. Thus, we argue that enumerability is a soft constraint and can easily be achieved in practice.
>
> **Prior works consider OC-space**
> We are unaware of any work explicitly identifying the OC space as an object that can be precomputed, stored in a data structure, and repeatedly queried in various ways. Earlier work on verification often generates and checks boxes that, from our viewpoint, are elements of the OC-space. However, none of these works conceptually define the OC space. We would greatly appreciate if the reviewer could point us to any relevant literature.
>
> **Verification is straightforward**
> Yes, simplicity is exactly the strength of our approach. Given a new property to verify, our approach allows them to easily write a algorithm that does this verification. This is an advantage.
>
> **Experimental results**
> The experiments aimed to demonstrate the feasibility of our generic approach by showing that it works over many datasets and verification properties, including entirely new properties. That it offers similar or better performance than existing special-purpose methods (i.e., targeting one task), is a nice bonus, but the value of our method does not depend on it, which is why we felt the experiments sufficed to make our point.
>
> From the review, it isn’t clear what precisely should be added to the paper apart from 2 extra datasets.
>
> **Q1: How were datasets chosen + Higgs/FashionMnist**
> We used the same datasets from Devos et al. ICML'25. Dataset size is mostly irrelevant: our method’s complexity depends on the size of the OC-space, which relates to the size of the model, not the dataset.
>
> We ran our method on FashionMNIST and Higgs. We could enumerate 53 out of 55 models on the Pareto front for FashionMNIST. For Higgs, we could enumerate 30 models out of 240 because the compression method (LOP) is slow; it often times out after 12h, returning partially compressed models which are too big to enumerate. All 30 enumerated models had an accuracy within 3% of the best XGBoost model. The largest enumerable model had 12.8M OCs and took 25s to enumerate. See Q3 for verification results.
>
> We also tested our method on an expected goals model, a widely used soccer metric. The XGBoost model has 1700 trees, 26234 leaves and a Brier score close to the best commercial models. After compression, the Brier score increased from 0.077 to 0.0789. The compressed model (30 trees, 669 leaves) has 150M OCs, enumerated in 989 seconds. This shows feasibility on a real-world task.
>
> **Q2: baselines chosen & implemented; solver configuration**
> We used well-established, publicly available implementations. We compare against exact baselines because our method is exact.
>
> * Robustness: Kantchelian’s MILP approach is the exact reference method. We use the public MILP implementation of Veritas and use the same configuration for Gurobi.
> * Fairness: we only know of 2 implementations: Veritas (Devos et al. ICML'21) and SMT (Devos et al. SDM'21). Veritas is exact in this setting and can enumerate all unfair boxes
> * Hybrid distance metric: we compare to Devos et al. SDM'21, which introduced this setting; we aren’t aware of other implementations. We use the original paper’s implementation of the paper and the same configuration for Z3
> * The average slope is novel; there are no competitors.
>
> **Q3:**
> Below, we extend Table 5 with Veritas + the FashionMnist and Higgs datasets and show the geometric mean of the runtime in seconds. On average, our exact approach is faster than Veritas except on Higgs. As an approximate approach, Veritas scales better to larger ensemble sizes.
>
> | |Kantchelian|Linear Scan|Rootbox Index|OCTree Index|Veritas|
> |:--|--:|--:|--:|--:|--:|
> |Adult|0.7153|0.0696|0.039|0.1515|0.055|
> |California|1.2407|0.2376|0.0208|0.1687|0.0646|
> |Credit|0.3418|0.0018|0.0056|0.0027|0.0435|
> |DryBean[6vRest]|0.3559|0.0014|0.0052|0.0008|0.0428|
> |Phoneme|2.4504|0.0482|0.0265|0.0467|0.0893|
> |Spambase|0.5545|0.1327|0.0326|0.0493|0.1105|
> |Vehicle|0.6287|0.0366|0.0186|0.017|0.0663|
> |Volkert[2v7]|0.8319|0.4885|0.0697|0.1927|0.2579|
> |FashionMnist[1vRest]|0.5448|0.4366|0.1118|0.1145|0.9803|
> |Higgs|1.3916|0.8274|0.1256|0.5132|0.0745|
> |Overall|0.7524|0.0665|0.0286|0.0436|0.098|

---

> > ### Author Rebuttal · Reviewer_Nd9T · 2026-04-04
> >
> > Hi,
> >
> > Thank you for your detailed response.
> >
> > - "practitioners may want to tradeoff performance for enumerability so that verification becomes feasible": I think this argument is pretty convincing, and would encourage the authors to feature it prominently in the paper.
> >
> > - Regarding prior work, **I believe it is a blatant misrepresentation to claim "[no prior works] conceptually define the OC space" when [1] for instance, which is one of the works you cited from 2023, most certainly defines OC-Space in Section 2.** As I stated in my review, I do see a lot of value in proposing the unifying perspective that you do propose in the paper; however, it you could be great if you could defend/clarify your novelty claims.
> >
> > - Thank you for the additional experiments and accompanying details! I do think they help paint a more complete picture, and would recommend incorporating it in the final manuscript.
> >
> > All things considered, I do think this is an interesting paper overall and in light of the added discussion will update my score to a 3.
> >
> > [1] Laurens Devos, Lorenzo Perini, Wannes Meert, Jesse Davis: Detecting Evasion Attacks in Deployed Tree Ensembles. ECML/PKDD (5) 2023: 120-136

---

> > > ### Author Response · Authors · 2026-04-08
> > >
> > > We apologize for the lack of precision and clarity in our wording. Our intention was to scope the claims about OC space in the context of existing tree ensemble verification algorithms as in "no prior verification methods conceptually defined OC space as a precomputed, stored, and repeatedly queried object." The novelty is explicitly precomputing, storing, and repeatedly using this data structure for verification. This contrasts with previous work:
> > >
> > > - Prior verification methods generate and test individual OC elements (e.g., boxes/regions), often implicitly operating in the OC space.
> > > - Devos et al. ECML'23 did define the OC space (as cited in the paper) but  (1) that work is not about verification and (2) it does not discuss precomputing the full OC space.

---

### Official Review · Reviewer_kPYq · 2026-03-13

**Soundness:** 3
**Presentation:** 3
**Significance:** 2
**Originality:** 2
**Overall Recommendation:** 5
**Confidence:** 4

**Summary:**

This paper addresses the challenge of verifying (ensembles of) decision trees. The authors propose a new algorithm that first enumerates all possible partitions of the input (the OC-space) and then searches through it. This neat separation allows for greater generality in verification algorithms and safety properties.

**Compliance With Llm Reviewing Policy:**

Affirmed.

**Final Justification:**

My recommendation is to accept the paper, subject to the authors including detailed pseudocode of their algorithm and full measumerents of the upfront cost of enumerating the OC space.

**Key Questions For Authors:**

1) Do the results in Figure 3-5 include the time required to build the OC-space or just the time required to search it? I am worried about the latter skewing the comparison in your favour.

**Limitations:**

A few words on the issue of enumerating large OC-spaces and which applications it might preclude would be appreciated.

**Strengths And Weaknesses:**

_Soundness._ The problem setting and algorithms proposed by the authors seem sound. Unfortunately, the lack of pseudocode prevents a thorough check of the finer details about searching the OC-space for each property. The experimental results support the algorithmic claims, except for the question below.

_Presentation._ There is no pseudocode on how to use the OC-space to verify the properties in Table 1. A reader interested in reproducing the paper, i.e. implementing the algorithms from scratch, is left with only the textual description as a guideline. It would be good to add such pseudocode as an appendix.

A minor comment. At the beginning of Section 3, the $\Pi$ notation is used to denote cartesian products of intervals. This is confusing, and I would encourage the authors to use the more common notation $\times$.

_Significance._ Enumerating the whole OC-space in advance might prevent the authors' algorithm to be used for very large trees. I can see how this could lead to exciting new research directions.

_Originality._ The unifying view that OC-spaces bring to decision tree verification seems novel.

---

> ### Author Rebuttal · Authors · 2026-03-31
>
> Dear reviewer, thank you for bringing up some insightful comments and questions. We will adapt the paper according to your feedback. Below, we address your comments.
>
> **We very much appreciated your comment that "I can see how this could lead to exciting new research directions.**
>
> Indeed, we believe the originality and significance lie in the generality of the OC-space view: it enables a wide variety of queries, including some for which no methods have been proposed before such as the "average slope" constraint and the OC-level robustness in Table 1 (and presumably many more that have not yet been considered in the literature).
>
> **On the lack of pseudocode on how to use the OC-space to verify the properties in Table 1**
>
> Our intention was for the formulas in Table 1 to serve as pseudocode as these can be directly translated to Python code. We agree pseudocode would also be helpful and will add it to the appendix for all properties considered in Table 1 (Theorems B1 and B2 give pseudocode for linear scans). To illustrate, computing notion (2) of empirical robustness (without an index) can be done using the following pseudocode:
>
>     given: example x, OC-space O
>     closest_dist = inf
>     foreach b in O:
>         if (dist(x,b)< closest_dist) and y \neq y':
>             closest_dist = mindist(x,b)
>     return closed_dist
>
> As another example, the code for "part of X where model is unfair" given an OCTreeBin index O and protected feature p, can be written in Python as:
>
>     results = []
>     for b in filter(dep(p), O.enumerate(O.negidx)):
>         for b2 in O.enumerate_overlapping(relax(b,p),O.posidx):
>             results.append(intersection(b,b2))
>
> where b, b', p map to b, b2 and p in the Python code. Generating b from the negative boxes and b' from the positive ensures y!=y'; the filter dep(p) lets only p-dependent b pass, and enumerate_overlapping enumerates those b' from positive boxes that overlap with relax(b,p).  The enumerate_overlapping function is part of the index functionality: generally, the index can generate boxes at distance <= c from a given box efficiently (more efficiently than generating all and filtering them afterwards), this is exactly the reason for the existence of spatial indexes such as KD-trees, X-trees, etc.  Overlapping boxes are a special case of this for c=0.
>
> **Q1: Do the results in Figure 3-5 include the time required to build the OC-space or just the time required to search it?**
>
> No, enumeration and index construction time is not included; this is stated in the sentence right before section 7.1. That time is relevant when only a single query is needed, but gets amortized per query when many queries are run afterwards. Our assumption is that queries such as robustness will be run many times, e.g., robustness may be checked for each prediction made with the model or on a large test set prior to model deployment. Similarly, one may want to look at multiple tasks on one model (e.g., robustness, fairness, and average slope).
>
> The table below shows the average number of queries one has to answer to make the cumulative runtime (i.e., including verification, enumeration, and index building) for our approach less than us the MILP solver for determining adversarial robustness wrt l infinity norm as discussed in section 7.2.  The overhead is usually paid off after a few hundred to a few thousand queries.
>
> |Dataset|Queries to Amortize (avg)|
> |:--|--:|
> |Credit|136|
> |Adult|590|
> |Vehicle|2429|
> |Spambase|3393|
> |Phoneme|595|
> |DryBean|412|
> |Volkert|1431|
> |California|1592|
>
> We would like to highlight the following points:
> * Indexes are applicable across multiple query types.  E.g., the results in Figures 3-5 for robustness with l_infinity norm, fairness, and robustness with a hybrid norm all use the same indexes
>
> * 95% of enumerations take < 1 minute and 98.8% in < 5 minutes
> * 95% of the rootbox indexes can be build in < 1 minute and 99.3% in < 5 minutes
> * 93% of the OC indexes can be build in < 1 minute and 97.7% in < 5 minutes
> * The implementations for constructing the indexes are not optimized
>
> The below table summarizes the geometric mean of the time (range of times is given in parenthesis) to enumerate, construct the rootbox index, and construct the OC index per dataset with all times in seconds:
>
> |Dataset|Enumeration Time (s)|Rootbox Index Time (s)|OCTree Index Time (s)|
> |:--|:--|:--|:--|
> |Credit|0.2275 (0.0352-1.8)|0.0118 (0.0042-1.29)|0.0815 (0.0040-7.74)|
> |Adult|1.0868 (0.1736-305)|0.1141 (0.0066-225)|2.2896 (0.0063-92466)|
> |Vehicle|0.8097 (0.3481-117)|0.1029 (0.0051-284)|0.3994 (0.0044-222)|
> |Spambase|1.3350 (0.3235-99)|0.3984 (0.0065-1184)|2.4286 (0.0064-797)|
> |Phoneme|0.7068 (0.1232-1908)|0.1088 (0.0055-238)|0.5358 (0.0050-449)|
> |DryBean|0.2958 (0.0984-1.63)|0.0086 (0.0052-0.012)|0.0157 (0.0047-5.24)|
> |Volkert|2.6383 (0.4581-288)|0.3479 (0.0059-541)|5.5875 (0.0064-901)|
> |California|1.2721 (0.1281-390)|0.2334 (0.0050-177)|5.1935 (0.3426-48263)|
>
> We will add this to the paper.

---

> > ### Author Rebuttal · Reviewer_kPYq · 2026-04-01
> >
> > Thanks for your response.
> >
> > i appreciated the additional table showing the number of queries required to amortise the cost of enumerating OC queries. It would definitely be a useful addition to the paper.
> >
> > Accordingly, I am going to increase my score to 5 - accept.

---

### Official Review · Reviewer_CKNm · 2026-03-15

**Soundness:** 4
**Presentation:** 4
**Significance:** 3
**Originality:** 3
**Overall Recommendation:** 5
**Confidence:** 4

**Summary:**

In this paper, the authors propose using the set of all possible output configurations as a generic approach for verification of tree ensembles. Since the ensemble prediction is the same inside each OC box, the authors re-formulate verification problems as searches over the OC-space as opposed to direct reasoning over the original piecewise function. Specifically they cover: plausibility of range, local robustness, fairness, monotonicity, safety, and multiplicity, and they additionally introduce OC-level robustness and an average-slope query. They then propose OC-space indexes: rootbox and OC-Tree(OC-TreeBin) based on sub-ensembles, and evaluate feasibility of enumerating OC-space and query-time speed against task-specific baselines on compressed XGBoost models.

**Compliance With Llm Reviewing Policy:**

Affirmed.

**Key Questions For Authors:**

-	What are the results for when OC enumeration and index-build time are included in end-to-end comparisons for one-off queries rather than repeated-query settings?
-	How essential is model compression to the practical feasibility claim?

**Limitations:**

No. I believe limitations are not discussed enough. The paper could discuss when enumeration becomes impractical, the memory burden, the dependence on compression and bounded domains for some queries The impact statement is also almost entirely positive.

**Strengths And Weaknesses:**

Soundness:
I consider the set-up to be technically sound: in every OC the output is constant, so verification essentially becomes search over configurations. W.r.t theoretical analysis, the authors present succintly all preliminaries, prove the theorems on the upper bounds of OC space size, and describe in detail the algorithmic approaches for all considered properties for verification. The authors experimented over a wide range of datasets and model configurations, and chose relevant baselines for the queried tasks. The only note I would make is that since the reported query times exclude OC-space enumeration and index construction, the runtime comparisons favor repeated-query settings more than one-off verification settings.


Presentation:
In my opinion, the paper is well-structured and written. The definitions are clear, the three search patterns are a useful organizing principle, and Table 1 is very useful in presenting an overview of all properties and algorithms. However, I feel that novelty relative to prior OC-like search methods could be stated more clearly. The authors do state that several existing approaches already operate in something “akin to” OC-space, so the real distinction is the explicit unifying view, full-space enumeration, and the proposed indexing strategy. This should be stated also in the introduction. Code is only promised upon acceptance.


Significance:
Through this contribution, the authors provide a way to move from previously separate verification problems into one reusable backend. I consider that overall, the approach could influence follow-up work, as the authors suggest future verifiers for tree ensembles to be designed around shared OC-space infrastructure rather than per-property algorithms. In particular, the paper could be influential for tree ensemble models. However, I think the practical impact could be affected by the need to enumerate and store OC-space first. All in all, the paper can likely advance a meaningful subarea and may shape future work there, but the evidence for broad everyday applicability is limited by the preprocessing necessities.

Originality:
I consider the originality to be good. Specifically, due to the explicit unifying OC-space approach, the sub-ensemble-based index construction, and the addition of new queries. At the same time, the paper itself notes that prior methods already reason in something close to OC-space, so the novelty is more about synthesis, generalization, and indexing.

---

> ### Author Rebuttal · Authors · 2026-03-31
>
> Dear reviewer, thank you for your extensive feedback. We appreciate the positive comments and especially your statement that this paper could be influential in the field of tree ensemble models. We share your view that this paper could influence follow-up work and hope to bring a novel perspective to verification of tree ensembles.
>
> Below are responses to your questions & comments.
>
> **Novelty relative to prior OC-like search methods could be stated more clearly**
>
> We are unaware of any work explicitly identifying the OC space as an object that can be precomputed, stored in a data structure, and repeatedly queried in various ways, which is what makes our approach so generic. Earlier work on verification often generates and checks boxes that, from our viewpoint, are elements of the OC-space. However, none of these works conceptually define the set of all such boxes. We will clear up this distinction in the introduction.
>
> **Bounded instance spaces**
>
> Our approach does not require bounded instance spaces. This statement was only made in relation to queries that return (relative) volumes: these queries are simply not meaningful for unbounded instance spaces because the volume inside an unbounded box is then infinite, so ratios of volumes may be infinite, zero or undefined.
>
> **Memory burden**
>
> We will discuss this in greater detail in the paper as we agree it is interesting and relevant. The amount of memory used depends on a wide variety of factors including the dimensionality of the data, the complexity of the ensemble, and how the structure is laid out (e.g., single vs. double precision, sparse structures or dense, ...).  We did not optimize for this (except for model compression). In our experiments, enumerating OCs up to 100 million is feasible on a MacBook Pro 2022 with 24G memory; beyond 100 million, one may need disk storage. Although disk storage is slower than memory, write and read operations to disk can be highly optimized due to efficient index partitioning.
>
> **Q1:**
> Indeed, our assumption is that queries such as robustness will be run many times, e.g., robustness may be checked for each prediction made with the model or on a large test set prior to model deployment. The table below shows after how many queries the time for enumeration + index construction is compensated by the lower verification times (compared to the MILP solver) for computing adversarial robustness wrt the l infinity norm as discussed in section 7.2.  The overhead is usually paid off after a few hundred to a few thousand queries.
>
> |Dataset|#Queries to Amortize (avg)|
> |:--|--:|
> |Credit|136|
> |Adult|590|
> |Vehicle|2429|
> |Spambase|3393|
> |Phoneme|595|
> |DryBean|412|
> |Volkert|1431|
> |California|1592|
>
> We would like to highlight the following points:
> * Indexes are applicable across multiple query types.  E.g., the results in Figures 3-5 for robustness with l_infinity norm, fairness, and robustness with a hybrid norm all use the same indexes.
>
> * 95% of enumerations take < 1 minute and 98.8% in < 5 minutes
>
> * 95% of the rootbox indexes can be build in < 1 minute and 99.3% in < 5 minutes
>
> * 93% of the OC indexes can be build in < 1 minute and 97.7% in < 5 minutes
>
> * The implementations for constructing the indexes are not optimized
>
> The below table summarizes the geometric mean of the time (range of times is given in parenthesis) to enumerate, construct the rootbox index, and construct the OC index per dataset with all times in seconds:
>
> |Dataset|Enumeration (s)|Rootbox index (s)|OCTree index (s)|
> |:--|:--|:--|:--|
> |Credit|0.2275 (0.0352-1.8)|0.0118 (0.0042-1.29)|0.0815 (0.0040-7.74)|
> |Adult|1.0868 (0.1736-305)|0.1141 (0.0066-225)|2.2896 (0.0063-92466)|
> |Vehicle|0.8097 (0.3481-117)|0.1029 (0.0051-284)|0.3994 (0.0044-222)|
> |Spambase|1.3350 (0.3235-99)|0.3984 (0.0065-1184)|2.4286 (0.0064-797)|
> |Phoneme|0.7068 (0.1232-1908)|0.1088 (0.0055-238)|0.5358 (0.0050-449)|
> |DryBean|0.2958 (0.0984-1.63)|0.0086 (0.0052-0.012)|0.0157 (0.0047-5.24)|
> |Volkert|2.6383 (0.4581-288)|0.3479 (0.0059-541)|5.5875 (0.0064-901)|
> |California|1.2721 (0.1281-390)|0.2334 (0.0050-177)|5.1935 (0.3426-48263)|
>
> **Q2:**
> In most cases, model compression is important because standard learners such as XGBoost have no incentive to learn small ensembles, even if small ensembles exist with the same accuracy as those returned by XGBoost. Most ensemble compression methods allow the user to trade off size versus accuracy. For the LOP method we used, it was shown in earlier work that it often achieves significant compression at 0.5% accuracy loss. The acceptable loss could be increased to 1% or more when necessary to find sufficiently small ensembles.
>
> Another way to view this is: one may want to trade off enumerability vs. performance because enumerability makes verification feasible. This is the same motivation used with interpretability: sometimes it is beneficial to sacrifice (a small amount of) accuracy to yield more interpretable models. We'll discuss this in the paper.

---

> > ### Author Rebuttal · Reviewer_CKNm · 2026-04-05
> >
> > Thanks for the rebuttal. I keep the score as it is.

---

> > > ### Author Response · Authors · 2026-04-08
> > >
> > > Thanks for your response. We were wondering in what other ways not addressed in the rebuttal our paper can be improved?

---

### Decision · Program_Chairs · 2026-04-30

**Decision:**

Accept (spotlight)

**Comment:**

The paper received three positive reviews with high confidence (accept x2, weak accept). One more negative reviewers (weak reject) viewed the paper as interesting. This seems to be a good paper that should be included in the program.